# Simple Yet Efficient Locality Sensitive Hashing with Theoretical Guarantee

## Abstract

Locality-sensitive hashing (LSH) is an effective randomized technique widely used in many machine learning tasks such as outlier detection, neural network training and nearest neighbor search. The cost of hashing is the main performance bottleneck of these applications because the index construction functionality, a core component dominating the end-to-end latency, involves the evaluation of a large number of hash functions. Surprisingly, however, little work has been done to improve the efficiency of LSH computation. In this paper, we design a simple yet efficient LSH scheme, named FastLSH, by combining random sampling and random projection. FastLSH reduces the hashing complexity from $O(n)$ to $O(m)$ ($m < n$), where $n$ is the data dimensionality and $m$ is the number of sampled dimensions. More importantly, FastLSH has provable LSH property, which distinguishes it from the non-LSH fast sketches. To demonstrate its broad applicability, we conduct comprehensive experiments over three machine learning tasks, i.e., outlier detection, neural network training and nearest neighbor search. Experimental results show that algorithms powered by FastLSH provides up to 6.1x, 1.7x and 20x end-to-end speedup in anomaly detection latency, training time and index construction, respectively. The source code is available at https://anonymous.4open.science/r/FastLSHForMachineLearning-7CAC.

## 1 Introduction

Locality-sensitive hashing (LSH) is an effective randomized technique in machine learning, which is originally proposed to solve the problem of approximate nearest neighbor (ANN) search in high dimensional space Indyk & Motwani (1998); Datar et al. (2004); Andoni & Indyk (2008). The basic idea of LSH is to map high dimensional points into buckets in low dimensional space using random hash functions, by which similar points have higher probability to end up in the same bucket than dissimilar points.

The LSH scheme for $l_2$ norm (E2LSH) is proposed in Datar et al. (2004); Andoni (2005) based on $p$-stable distributions. Owing to the sub-linear time complexity and theoretical guarantee on query accuracy, E2LSH is arguably one of the most popular ANN search algorithms both in theory and practice. Many variants of E2LSH have been proposed to achieve much better space occupation and query response time Lv et al. (2007); Tao et al. (2010); Gan et al. (2012); Sun et al. (2014); Huang et al. (2015); Lu & Kudo (2020); Yang et al. (2020); Zheng et al. (2020); Tian et al. (2022). Throughout this article, we focus on the LSH scheme under $l_2$ norm, and the extension to angular similarity, maximum inner product and $l_p$ norm ($p \in (0, 2)$) is discussed in Section 5.

In addition to answering ANN queries, LSH finds applications in many other domains. To name a few, Arrays of (locality-sensitive) Count Estimators (ACE) detects anomaly in data by performing lookup in hash tables, where the counts of collision are used as estimators for outlier Luo & Shrivastava (2018). ACE first processes high-dimensional data using LSH and constructs multiple hash tables. Each hash bucket is equipped with a counter to record how many data points fall into that bucket. ACE then leverages these counters across multiple hash tables to analyze the collision frequency of a given query. If a query shows a low collision frequency, it may be flagged as an outlier. Although ACE efficiently uses collisions to detect outliers, building the hash tables and counters is quite time-consuming.

SLIDE (Sub-LInear Deep learning Engine), a novel deep learning engine, uses LSH and parallel programming to achieve more efficient training on large-scale recommendation datasets using solely multi-core CPUs Chen et al. (2020). SLIDE maps the weight vectors of neurons in each layer of the neural network to be hash codes and then store these neurons into hash tables. During the forward propagation, SLIDE can quickly retrieve and activate the relevant neurons from these hash buckets based on the hash codes of the input data (acted as query). However, building these hash tables and updating them due to weight changes during the backward propagation often takes significantly more time than simply retrieving the neurons. Especially when neuron weights undergo substantial changes, the reconstruction and update of hash tables can greatly increase computational overhead.

For aforementioned LSH-based applications, the nearest neighbor search is a part of a larger application, where index construction time is more important than query time. This is because that the LSH index has to be built frequently due to data update or gradient update (SLIDE) or the index construction time occupies a large proportion of the end-to-end execution time (ACE). The index construction time translates directly to hashing cost because building an index involves massive computation of LSH functions. Take the widely used E2LSH as an example, computing $k$ hashes of a vector $v$ takes $O(nk)$ computation, where $n$ is the dimensionality of $v$ and $k$ is the number of hash functions. For typical ANN search task, $k$ commonly ranges from few hundreds to several thousands, and keeps growing with the cardinality of the dataset ($N$) since the number of hashes required by E2LSH is $O(N^\rho)$ Datar et al. (2004). To sum up, hashing cost (index construction time) is the main computational and resource bottleneck step in almost such LSH-based applications, especially when data come in a streaming fashion and/or LSH data structures have to be constructed repeatedly Yang et al. (2020); Sundaram et al. (2013).

Surprisingly enough, little endeavor has been made on more efficient LSH schemes under $l_2$ norm. The only known technique, termed as ACHash Dasgupta et al. (2011), exploits fast Hadamard transform to estimate the distance distortion of two points in the Euclidean space. This method, like other fast JL sketches Ailon & Chazelle (2006); Ailon & Liberty (2009), does not owns the provable LSH property. Thus, it is not a desirable alternative to the standard LSH because there are substantial empirical evidence that using these (non-LSH) sketches incurs a drastic bias in the expected behavior, leading to poor accuracy Shrivastava (2017).

**Our Contribution:** We develop a simple yet efficient LSH scheme (FastLSH), which needs only two basic operations – random sampling and random projection, and offers better time complexity than E2LSH. Also, we derive the expression of the probability of collision (equity of hash values) for FastLSH and prove the asymptotic equivalence between FastLSH and E2LSH, which means that our proposal owns the desirable LSH property. We also rigidly analyze how the sampling ratio affects the probability of collision when the number of sampled dimensions is relatively small. To further validate our claims, we conduct comprehensive experiments over three machine learning tasks, i.e., outlier detection, neural network training and nearest neighbor search, in which the standard LSH is replaced by FastLSH. Experimental results show that algorithms powered by FastLSH provide significant end-to-end speedup in anomaly detection latency, training time and the index construction time, respectively.

## 2 PRELIMIANRIES

In this section, we introduce notations and background knowledge used in this article. Let $\mathbf{D}$ be the dataset of size $N$ in $\mathbb{R}^n$ and $v \in \mathbf{D}$ be a data point (vector) and $u \in \mathbb{R}^n$ be a query vector. We denote $\phi(x) = \frac{1}{\sqrt{2\pi}} \exp(-\frac{x^2}{2})$ and $\Phi(x) = \int_{-\infty}^{x} \frac{1}{\sqrt{2\pi}} \exp(-\frac{x^2}{2})dx$ as the probability density function (PDF) and cumulative distribution function (CDF) of the standard normal distribution $\mathcal{N}(0, 1)$, respectively.

### 2.1 LOCALITY SENSITIVE HASHING

**Definition 2.1.** (Locality Sensitive Hashing) A hash function family $\mathcal{H} = \{h : \mathbb{R}^n \to U\}$ is called $(R, cR, p_1, p_2)$-sensitive if for any $v, u \in \mathbb{R}^n$

- if $\|v - u\|_2 \leq R$ then $Pr_{\mathcal{H}}[h(v) = h(u)] \geq p_1$;

- if $\|\boldsymbol{v} - \boldsymbol{u}\|_2 \geq cR$ then $Pr_{\mathcal{H}}[h(\boldsymbol{v}) = h(\boldsymbol{u})] \leq p_2$;

In order for the LSH family to be useful, it has to satisfy $c > 1$ and $p_1 > p_2$. Please note that only hashing schemes with such a property are qualified locality sensitive hashing and can enjoy the theoretical guarantee of LSH.

Datar et al. (2004) presents an LSH family that can be employed for $l_p$ ($p \in (0, 2]$) norms based on $p$-stable distribution. When $p = 2$, it yields the well-known LSH family for $l_2$ norm (E2LSH). The hash function is defined as follows:

$$h_{\boldsymbol{a},b}(\boldsymbol{v}) = \left\lfloor \frac{\boldsymbol{a}^T \boldsymbol{v} + b}{w} \right\rfloor \tag{1}$$

where $\lfloor \rfloor$ is the floor operation, $\boldsymbol{a}$ is a $n$-dimensional vector with each entry chosen independently from $\mathcal{N}(0, 1)$ and $b$ is a real number chosen uniformly from the range $[0, w]$. $w$ is an important parameter by which one could tune the performance of E2LSH.

For E2LSH, the probability of collision of $(\boldsymbol{v}, \boldsymbol{u})$ under $h_{\boldsymbol{a},b}(\cdot)$ is computed as

$$p(s) = Pr[h_{\boldsymbol{a},b}(\boldsymbol{v}) = h_{\boldsymbol{a},b}(\boldsymbol{u})] = \int_0^w f_{|sX|}(t)(1 - \frac{t}{w})dt \tag{2}$$

where $s = \|\boldsymbol{v} - \boldsymbol{u}\|_2$ is the Euclidean distance between $(\boldsymbol{v}, \boldsymbol{u})$, and $f_{|sX|}(t)$ is the PDF of the absolute value of normal distribution $sX$ ($X$ is a random variable following the standard normal distribution). Given $w$, $p(s)$ is a monotonically decreasing function of $s$, which means $h_{\boldsymbol{a},b}(\cdot)$ satisfies the LSH property.

## 2.2 TRUNCATED NORMAL DISTRIBUTION

The truncated normal distribution is suggested if one need to use the normal distribution to describe the random variation of a quantity that, for physical reasons, must be strictly in the range of a truncated interval instead of $(-\infty, +\infty)$ Cohen (1991). The truncated normal distribution is the probability distribution derived from that of normal random variables by bounding the values from either below or above (or both). Assume that the interval $(a_1, a_2)$ is the truncated interval, then the probability density function can be written as:

$$\psi(x; \mu, \sigma^2, a_1, a_2) = \begin{cases} 0 & x \leq a_1 \\ \frac{\phi(x; \mu, \sigma^2)}{\Phi(a_2; \mu, \sigma^2) - \Phi(a_1; \mu, \sigma^2)} & a_1 < x < a_2 \\ 0 & a_2 \leq x \end{cases} \tag{3}$$

The cumulative distribution function is:

$$\Psi(x; \mu, \sigma^2, a_1, a_2) = \begin{cases} 0 & x \leq a_1 \\ \frac{\Phi(x; \mu, \sigma^2) - \Phi(a_1; \mu, \sigma^2)}{\Phi(a_2; \mu, \sigma^2) - \Phi(a_1; \mu, \sigma^2)} & a_1 < x < a_2 \\ 1 & a_2 \leq x \end{cases} \tag{4}$$

## 3 FAST LSH VIA RANDOM SAMPLING

### 3.1 THE PROPOSED LSH FUNCTION FAMILY

The cost of hashing defined in Eqn. 1 is dominated by the inner product $\boldsymbol{a}^T \boldsymbol{v}$, which takes $O(n)$ multiplication and addition operations. As mentioned in Section 1, hashing is one of the main computational bottleneck in almost all LSH-based applications. To address this issue, we propose a novel family of locality sensitive hashing termed as FastLSH. Computing hash values with FastLSH involves two simple steps, i.e., random sampling and random projection.

In the first step, we do random sampling from $n$ dimensions. Particularly, we draw $m$ *i.i.d.* samples in the range of 1 to $n$ uniformly to form a multiset $S$. For every $\boldsymbol{v} = \{v_1, v_2, \cdots, v_n\}$, we concatenate all $v_i$ to form a $m$-dimensional vector $\tilde{\boldsymbol{v}} = \{\tilde{v}_1, \tilde{v}_2, \cdots, \tilde{v}_m\}$ if $i \in S$. As a quick example, suppose $\boldsymbol{v} = \{1, 3, 5, 7, 9\}$ is a 5-dimensional vector and $S = \{2, 4, 2\}$. Then we can get a 3-dimensional vector $\tilde{\boldsymbol{v}} = \{3, 7, 3\}$ under $S$. It is easy to see that each entry in $\boldsymbol{v}$ has equal

probability $\frac{m}{n}$ of being chosen. Next, we slightly overuse notation $S$ and denote by $S(\cdot)$ the random sampling operator, that is, $\tilde{\boldsymbol{v}} \in \mathbb{R}^m = S(\boldsymbol{v})$ for $\boldsymbol{v} \in \mathbb{R}^n$ $(m < n)$.

In the second step, the hash value is computed in the same way as Eqn. 1 using $\tilde{\boldsymbol{v}}$ instead of $\boldsymbol{v}$, and then the overall hash function is formally defined as follows:

$$h_{\tilde{\boldsymbol{a}}, \tilde{b}}(\boldsymbol{v}) = \left\lfloor \frac{\tilde{\boldsymbol{a}}^T S(\boldsymbol{v}) + \tilde{b}}{\tilde{w}} \right\rfloor \tag{5}$$

where $\tilde{\boldsymbol{a}} \in \mathbb{R}^m$ is the random projection vector of which each entry is chosen independently from $\mathcal{N}(0, 1)$, $\tilde{w}$ is a user-specified constant and $\tilde{b}$ is a real number uniformly drawn from $[0, \tilde{w}]$. The hash function $h_{\tilde{\boldsymbol{a}}, \tilde{b}}(\boldsymbol{v})$ maps a $n$-dimensional vector $\boldsymbol{v}$ onto the set of integers.

Compared with E2LSH, FastLSH reduces the complexity of hashing from $O(n)$ to $O(m)$. As will be discussed in Section 6, a relatively small $m < n$ suffices to provide competitive performance against E2LSH, which leads to significant performance gain in hash function evaluation.

## 4 THEORETICAL ANALYSIS

While FastLSH is easy to comprehend and simple to implement, it is non-trivial to show that the proposed LSH function meets the LSH property, i.e., the probability of collision for $(\boldsymbol{v}, \boldsymbol{u})$ decreases as their $l_2$ distance increases. In this section, we first derive the probability of collision for FastLSH in Theorem 4.2, and then prove that its asymptotic behavior is equivalent to E2LSH in Corollary 4.7 and Fact 4.5. Thereafter, by using both rigid analysis and numerical method, we demonstrate that FastLSH still owns desirable LSH property even if $m$ is relatively small in Lemma 4.8 and Fact 4.9.

### 4.1 PROBABILITY OF COLLISION

For given vector pair $(\boldsymbol{v}, \boldsymbol{u})$, let $s = \|\boldsymbol{v} - \boldsymbol{u}\|_2$. The collection of $n$ entries $(v_i - u_i)^2$ $\{i = 1, 2, \ldots, n\}$ follows an unknown distribution with a finite mean $\mu = (\sum_{i=1}^n (v_i - u_i)^2)/n$ and variance $\sigma^2 = (\sum_{i=1}^n ((v_i - u_i)^2 - \mu)^2)/n$. After performing the sampling operator $S(\cdot)$ of size $m$, $\boldsymbol{v}$ and $\boldsymbol{u}$ are transformed into $\tilde{\boldsymbol{v}} = S(\boldsymbol{v})$ and $\tilde{\boldsymbol{u}} = S(\boldsymbol{u})$, and the squared distance of $(\tilde{\boldsymbol{v}}, \tilde{\boldsymbol{u}})$ is $\tilde{s}^2 = \sum_{i=1}^m (\tilde{v}_i - \tilde{u}_i)^2$. By Central Limit Theorem, we have the following lemma:

**Lemma 4.1.** *If $m$ is sufficiently large, then the sum $\tilde{s}^2$ of $m$ i.i.d. random samples $(\tilde{v}_i - \tilde{u}_i)^2$ $(i \in 1, 2, \ldots, m)$ converges asymptotically to the normal distribution with mean $m\mu$ and variance $m\sigma^2$, i.e., $\tilde{s}^2 \sim \mathcal{N}(m\mu, m\sigma^2)$.*

Lemma 4.1 states that the squared distance between $\tilde{\boldsymbol{v}}$ and $\tilde{\boldsymbol{u}}$ follows a normal distribution for large $m$. Practically, a small $m$ (say 30) often suffices to make the sampling distribution of the sample mean approaches the normal in real-life applications Islam (2018); Feller (1991).

Recall that $\boldsymbol{a}$ is a projection vector with entries being *i.i.d* samples drawn from $\mathcal{N}(0, 1)$. It follows from the $p$-stability that the distance between projections $(\boldsymbol{a}^T \boldsymbol{v} - \boldsymbol{a}^T \boldsymbol{u})$ for two vectors $\boldsymbol{v}$ and $\boldsymbol{u}$ is distributed as $\|\boldsymbol{v} - \boldsymbol{u}\|_2 X$, i.e., $sX$, where $X \sim \mathcal{N}(0, 1)$ Zolotarev (1986); Datar et al. (2004). Similarly, the projection distance between $\tilde{\boldsymbol{v}}$ and $\tilde{\boldsymbol{u}}$ under $\tilde{\boldsymbol{a}}$ $(\tilde{\boldsymbol{a}}^T \tilde{\boldsymbol{v}} - \tilde{\boldsymbol{a}}^T \tilde{\boldsymbol{u}})$ follows the distribution $\tilde{s}X$. Note that the PDF of $sX$, i.e., $f_{sX}(x) = \frac{1}{s}\phi(\frac{x}{s})$, is an important factor in calculating the probability of collision in Eqn. 2. Hence, if we know the PDF of $\tilde{s}X$ we can derive easily the probability of collision for vector pair $(\boldsymbol{v}, \boldsymbol{u})$ under the proposed LSH function.

Let $f_{|\tilde{s}X|}(t)$ represent the PDF of the absolute value of $\tilde{s}X$. By replacing $f_{|sX|}(t)$ in Eqn. 2 with $f_{|\tilde{s}X|}(t)$, we have the collision probability $p(s, \sigma)$ for FastLSH as follows.

**Theorem 4.2.**

$$p(s, \sigma) = Pr[h_{\tilde{\boldsymbol{a}}, \tilde{b}}(\boldsymbol{v}) = h_{\tilde{\boldsymbol{a}}, \tilde{b}}(\boldsymbol{u})] = \int_0^{\tilde{w}} f_{|\tilde{s}X|}(t)(1 - \frac{t}{\tilde{w}})dt \tag{6}$$

*Proof.* See Appendix A.3 $\qquad\qquad\qquad\qquad\qquad\qquad\qquad\qquad\qquad\qquad\qquad\qquad\qquad\square$

Next we will show how to compute $f_{|\tilde{s}X|}(t)$. Note that random variable $\tilde{s}^2$ does not follow exactly the normal distribution since $\tilde{s}^2 \geq 0$ whereas the range of definition of the normal distribution is

$(-\infty, +\infty)$. A mathematically defensible way to preserve the main features of the normal distribution while avoiding negative values involves *the truncated normal distribution*, in which the range of definition is made finite at one or both ends of the interval.

Particularly, $\tilde{s}^2$ can be modeled by normal distribution $\tilde{s}^2 \sim \mathcal{N}(m\mu, m\sigma^2)$ over the truncation interval $[0, +\infty)$, that is, the singly-truncated normal distribution $\psi(x; \tilde{\mu}, \tilde{\sigma}^2, 0, +\infty)$. Considering the fact that $\tilde{s} \geq 0$, we have $Pr[\tilde{s} < t] = Pr[\tilde{s}^2 < t^2]$ for any $t > 0$. Therefore, the CDF of $\tilde{s}$, denoted by $F_{\tilde{s}}$, can be computed as follows:

$$F_{\tilde{s}}(t) = Pr[\tilde{s} < t] = Pr[\tilde{s}^2 < t^2] = \int_0^{t^2} \psi(x; \tilde{\mu}, \tilde{\sigma}^2, 0, \infty) dx \tag{7}$$

where $\tilde{\mu} = m\mu$ and $\tilde{\sigma}^2 = m\sigma^2$. Due to the fact that the PDF is the derivative of the CDF, the PDF of $\tilde{s}$, denoted by $f_{\tilde{s}}$, is derived as follows:

$$f_{\tilde{s}}(t) = \frac{d}{dt}[F_{\tilde{s}}(t)] = 2t\psi(t^2; \tilde{\mu}, \tilde{\sigma}^2, 0, \infty) \tag{8}$$

Although we know the distribution functions of both $\tilde{s}$ and $X$, it is not straight-forward to figure out the distribution of their product $\tilde{s}X$. Fortunately, Lemma 4.3 gives the characteristic function of random variable $W = XY$, where $X$ and $Y$ are two independent random variables, one following a standard normal distribution and the other following a distribution with mean $\mu$ and variance $\sigma^2$.

**Lemma 4.3.** *The characteristic function of the product of two independent random variables $W = XY$ is*

$$\varphi_W(x) = E_Y\{\exp(-\frac{x^2 Y^2}{2})\}$$

*where $X$ is a standard normal random variable and $Y$ is an independent random variable with mean $\mu$ and variance $\sigma^2$.*

*Proof.* See Appendix A.1 □

Note that the distribution of a random variable is determined uniquely by its characteristic function. As a result, the characteristic function of $\tilde{s}X$ can be obtained by Lemma 4.3 since $X$ follows the standard normal.

**Lemma 4.4.** *The characteristic function of $\tilde{s}X$ is*

$$\varphi_{\tilde{s}X}(x) = \frac{1}{2(1 - \Phi(\frac{-\tilde{\mu}}{\tilde{\sigma}}))} \exp(\frac{1}{8}x^4\tilde{\sigma}^2 - \frac{1}{2}\tilde{\mu}x^2) \operatorname{erfc}(\frac{\frac{1}{2}x^2\tilde{\sigma}^2 - \tilde{\mu}}{\sqrt{2}\tilde{\sigma}}) \quad (-\infty < x < +\infty)$$

*where $\operatorname{erfc}(t) = \frac{2}{\sqrt{\pi}} \int_t^{+\infty} \exp(-x^2)dx \; (-\infty < t < +\infty)$ is the complementary error function.*

*Proof.* See Appendix A.2 □

Given the characteristic function, the probability density function of $\tilde{s}X$ (denoted by $f_{\tilde{s}X}(t)$) can be obtained through the inverse Fourier transformation.

$$f_{\tilde{s}X}(t) = \frac{1}{2\pi} \int_{-\infty}^{+\infty} \exp(-itx)\varphi_{\tilde{s}X}(x)dx \tag{9}$$

where the symbol $i = \sqrt{-1}$ represents the imaginary unit.

## 4.2 THE ASYMPTOTIC BEHAVIOR OF FASTLSH

We can see that, unlike E2LSH, the probability of collision $p(s, \sigma)$ depends on both $s$ and $\sigma$. From this point of view, FastLSH can be regarded as a generalized version of E2LSH by considering one additional impact factor, i.e., the variation in the squared distance of each dimension for vector pair $(\boldsymbol{v}, \boldsymbol{u})$, making it more difficult to prove the LSH property.

From Eqn. 2 and Eqn. 6, one can see that the expressions of the probability of collision for E2LSH and FastLSH are quite similar. Actually, if $f_{\tilde{s}X}(t)$ follows normal distribution $\mathcal{N}(0, \frac{ms^2}{n})$, we can always make $p_w(s) = p_{\tilde{w}}(s, \sigma)$ by scaling $\tilde{w}$ to $\frac{mw}{n}$ based on Fact 4.5.

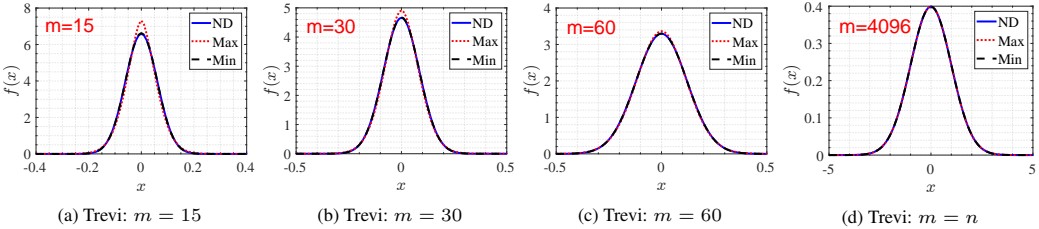

(a) Trevi: $m = 15$  (b) Trevi: $m = 30$  (c) Trevi: $m = 60$  (d) Trevi: $m = n$

Figure 1: Comparison of probability density curves of $\mathcal{N}(0, \frac{ms^2}{n})$ (ND) and $\tilde{s}X$ under different $m$ over *Trevi*.

**Fact 4.5** (Datar et al. (2004)). For E2LSH, $f_{sX}(t)$ follows the normal distribution $\mathcal{N}(0, s^2)$ and the collision probability $p(s)$ with bucket width $w$ is equal to $p(\alpha s)$ under the bucket width $\alpha w$, i.e., $p_w(s) = p_{\alpha w}(\alpha s)$ where $\alpha > 0$.

Proving the equivalence between $f_{\tilde{s}X}(t)$ and $\mathcal{N}(0, \frac{ms^2}{n})$ directly is not easy. To get around, the following theorem gives the asymptotic behavior of the characteristic function of $\tilde{s}X$.

**Theorem 4.6.**

$$\lim_{m \to +\infty} \frac{\varphi_{\tilde{s}X}(x)}{\exp(-\frac{ms^2 x^2}{2n})} = 1$$

where $\exp(-\frac{ms^2 x^2}{2n})$ is the characteristic function of $\mathcal{N}(0, \frac{ms^2}{n})$.

*Proof.* See Appendix A.4. □

Note that $\exp(-\frac{ms^2 x^2}{2n})$ is the characteristic function of $\mathcal{N}(0, \frac{ms^2}{n})$. As a result, Theorem 4.6 implies that $f_{\tilde{s}X}(t)$ is asymptotically identical to $\mathcal{N}(0, \frac{ms^2}{n})$ because a probability distribution is uniquely determined by its characteristic function, which immediately gives the following Corollary

**Corollary 4.7.** $f_{\tilde{s}X}(t) \sim$ *the PDF of* $\mathcal{N}(0, \frac{ms^2}{n})$ *as $m$ approaches infinity.*

By Corollary 4.7 and Fact 4.5, $p(s) = p(s, \sigma)$ asymptotically if $\tilde{w} = \frac{m}{n}w$, meaning that $\sigma$ has no effect on the probability of collision, and FastLSH is equivalent to E2LSH in this case.

### 4.3 The LSH Property for limited $m$

In practical scenarios, the number of sampled dimensions ($m$) is often limited. Next, we study the relation between FastLSH and E2LSH in this case by examining the similarity in $f_{\tilde{s}X}(t)$ and the PDF of $\mathcal{N}(0, \frac{ms^2}{n})$. Note that the similarity between $f_{\tilde{s}X}(t)$ and $\mathcal{N}(0, \frac{ms^2}{n})$ directly translates to the equivalence between $p(s)$ and $p(s, \sigma)$.

Since distributions near the normal can be decided very well given the first four moments Leslie (1959); Johnson (1949); Ramberg et al. (1979), we conduct the analysis by comparing the first four moments of corresponding distributions. The first four moments of $\tilde{s}X$ and $\mathcal{N}(0, \frac{ms^2}{n})$ are given in Lemma 4.8 and Fact 4.9, respectively.

**Lemma 4.8.**

$$\begin{cases} E(\tilde{s}X) & = 0 \\ E((\tilde{s}X)^2) = \frac{ms^2}{n}(1 + \epsilon) \\ E((\tilde{s}X)^3) = 0 \\ E((\tilde{s}X)^4) = \frac{3m^2 s^4}{n^2}(1 + \lambda) \end{cases}$$

*where* $\epsilon = \frac{\tilde{\sigma} \exp(\frac{-\tilde{\mu}^2}{2\tilde{\sigma}^2})}{\sqrt{2\pi}\tilde{\mu}(1 - \Phi(\frac{-\tilde{\mu}}{\tilde{\sigma}}))}$ *and* $\lambda = \frac{\tilde{\sigma}^2}{\tilde{\mu}^2} + \epsilon$.

*Proof.* See Appendix A.5 □

**Fact 4.9.** Hoel et al. (1971) The first four moments of $\mathcal{N}(0, \frac{ms^2}{n})$ are:

$$\begin{cases} E(sX) & = 0 \\ E((sX)^2) = \frac{ms^2}{n} \\ E((sX)^3) = 0 \\ E((sX)^4) = \frac{3m^2s^4}{n^2} \end{cases}$$

Lemma 4.8 and Fact 4.9 indicate that the first and third moments of $\tilde{s}X$ are equal to those of $\mathcal{N}(0, \frac{ms^2}{n})$, and $\tilde{s}X$ differ with $\mathcal{N}(0, \frac{ms^2}{n})$ in the second and fourth moments by only factors of $1 + \epsilon$ and $1 + \lambda$, respectively. It is easy to see that the smaller $\epsilon$ and $\lambda$ are, the closer $\tilde{s}X$ is to $\mathcal{N}(0, \frac{ms^2}{n})$.

Note that $\epsilon$ and $\lambda$ are monotonously decreasing (increasing) functions of $m$ ($\sigma$) since $\frac{\tilde{\sigma}}{\tilde{u}} = \frac{\sigma n}{\sqrt{ms^2}}$.

As a result, the first four moments of $\tilde{s}X$ and $\mathcal{N}(0, \frac{ms^2}{n})$ are equal with each other as $m$ approaches infinity because $\epsilon = \lambda = 0$ in this case. This analytical result is consistent with Corollary 4.7.

For limited $m < n$, the impact of $\sigma$ is not negligible. However, we can always adjust $m$ to control the impact of $\sigma$ (the data-dependent factor) on $f_{\tilde{s}X}(t)$ within a reasonable range. In this sense, Lemma 4.8 and Fact 4.9 provide a principled approach to quantitatively analyze how $m$ affects the difference between FastLSH and the classic LSH in terms of $\epsilon$ and $\lambda$. By using this analytical tool, it is easy for practitioners to determine the trade-off between hashing time (how much $m$ is) and desired performance level (how close FastLSH is to the standard LSH).

To visualize the similarity, we plot $f_{\tilde{s}X}(t)$ for different $m$ under the maximum and minimum $\sigma$, and the PDF of $\mathcal{N}(0, \frac{ms^2}{n})$ in Figure 1 on *Trevi*. More plots for other datasets are shown in Figure 7 in Appendix C.6 due to space limitation. Three observations can be made from these figures: (1) the distribution of $\tilde{s}X$ matches very well with $\mathcal{N}(0, \frac{ms^2}{n})$ for small $\sigma$; (2) for large $\sigma$, $f_{\tilde{s}X}(t)$ differs only slightly from $\mathcal{N}(0, \frac{ms^2}{n})$ for all $m$, implying that $s$ is the dominating factor in $p(s, \sigma)$; (3) greater $m$ results in higher similarity between $f_{\tilde{s}X}(t)$ and $\mathcal{N}(0, \frac{ms^2}{n})$, demonstrating that FastLSH can always achieve the same performance as E2LSH by choosing $m$ appropriately. The comparison of $\rho$, an important performance indicator for locality sensitive hashing, is reported in Appendix C.7 due to space limitation. Likewise, the comparison of $\rho$ suggests that FastLSH is equivalent to E2LSH even if in the case of limited $m$.

We further list the values of $\epsilon$ and $\lambda$ for different $m$ over 12 datasets in Table 10 in Appendix C.8 due to space constraints, where $\epsilon$ and $\lambda$ are calculated using the maximum, mean and minimum $\sigma$, respectively. As shown in Table 10, $\epsilon$ and $\lambda$ decrease as $m$ increases. Take *Trevi* as an example, $\epsilon$ is equal to 0 and $\lambda$ is very tiny (0.0001-0.000729), manifesting the equivalence between $f_{\tilde{s}X}(t)$ and $\mathcal{N}(0, \frac{ms^2}{n})$ for limited $m$.

## 5 EXTENSION TO OTHER SIMILARITY METRICS

In this section, we sketch how to extend FastLSH to other similarity measures. Since the angular similarity can be well approximated by the Euclidean distance if the norms of data item are identical, one can use FastLSH for the angular similarity directly after data normalization. In addition, FastLSH can solve the maximum inner product search problem by utilizing two transformation functions Bachrach et al. (2014). The detailed discussion is given in Appendix B.1. The extension of FastLSH to support $l_p$ norm for $p \in (0, 2)$ is worked, when we vary $l_2$ norm in Section 4 to $l_p$ norm and then with similar analysis, see Appendix B.2.

## 6 EXPERIMENTS

In this section, we conduct comprehensive experiments for three machine learning tasks, i.e., outlier detection, neural network training and nearest neighbor search to demonstrate the efficiency of our proposal. Due to space limitation, we only report the main results here and defer more information about datasets, parameter settings and additional experiments to Appendix C.1, C.2, C.3 and C.4.

## 6.1 HASH FUNCTIONS, BASELINES AND EVALUATION METRICS

**Hash Functions:** Three hash functions, i.e., E2LSH Datar et al. (2004), ACHash Dasgupta et al. (2011) and FastLSH, are compared. E2LSH is the classic LSH scheme for $l_2$ norm. ACHash is proposed to speedup the hash function evaluation by using Hadamard transform and sparse random projection[1]. It is worth noting that FastLSH can be easily plugged into any existing LSH applications considering its simplicity.

**Baselines:** For outlier detection, Arrays of (locality-sensitive) Count Estimators (ACE) Luo & Shrivastava (2018) is considered, which uses multiple LSH tables to estimate counts of collision and detect anomalies by performing lookup in hash tables. For neural network training, we examine SLIDE (Sub-LInear Deep learning Engine) Chen et al. (2020), a novel deep learning engine that combines smart randomized algorithms (LSH) and multi-core parallelism to achieve fast network training solely using CPU. For nearest neighbor search, two popular algorithms, i.e., E2LSH Datar et al. (2004); Andoni (2005) and MPLSH Lv et al. (2007) are evaluated. MPLSH is a variant of vanilla LSH (E2LSH), which provides better space and time efficiency. More information of these baselines are described in Appendix C.1, C.2, C.3 and C.4 respectively.

**Metrics:** For outlier detection, five performance measures i.e., outliers detected, correctly reported outliers, outlier missed, execution time and speedup, are listed. For neural network training task, we report the classification accuracy, the end-to-end training time and the number of iterations. For nearest neighbor search, we report the recall, i.e., the fraction of near neighbors that are correctly returned, the average query time, hashing cost and the index construction time.

## 6.2 DATASETS AND PARAMETER SETTINGS

**Outlier Detection:** For anomaly detection, we choose three real-world benchmark datasets, 1) Statlog Shuttle, 2) a9a and 3) Musk. The details of datasets for the anomaly detection task are described in AppendixC.1.

**Neural Network Training:** We employ two large real datasets, Delicious-200K and Amazon-670K, from the Extreme Classification Repository as in Bhatia et al. (2016). The statistics of datasets for neural network training are shown in AppendixC.2.

**Nearest Neighbor Search:** 11 publicly available high-dimensional real datasets and one synthetic dataset, i.e., Sun, Cifar, Audio, Trevi, Notre, Sift, Gist, Deep, Ukbench, Glove, ImageNet, Random, are experimented with Li et al. (2019). The details and statistics of which are presented in Appendix C.3.

Due to space limitation, please refer to Appendix C.1, C.2, C.3 and C.4 for more information about parameter settings.

## 6.3 RESULTS AND DISCUSSION

For three machine learning tasks, i.e., outlier detection, neural network training and nearest neighbor search, FastLSH can reduce the index construction time significantly and achieve almost the same or better recall and query time as LSH-based algorithms. Below, we present the experimental results for each task to validate this claim.

**Outlier Detection:** FastLSH can significantly reduce the end-to-end anomaly detection latency. The results for Musk are shown in Table 1. We can see that FastACE (FastLSH + ACE) offers around the same performance as ACE in terms of the numbers of correctly reported outliers and missed ones, whereas it needs much lower end-to-end execution time, achieving 6.1x and 2.6x speedup over ACE and ACHashACE (ACHash + ACE) thanks to the efficiency of FastLSH. ACHashACE missed more true anomaly because of the lack of theoretical guarantee.

Likewise, the results for *a9a* and *Statlog Shuttle*, as shown in Table 2 and Table 3, verify the superior performance of FastACE. One can see that FastACE delivers 4x and 1.2x speedup in the outlier

---

[1]Note that ACHash is actually not an eligible LSH method because no expression of the probability of collision exists for ACHash, not mentioning the desirable LSH property. We choose ACHash for the sake of completeness.

detection latency. For both *a9a* and *Statlog Shuttle*, FastACE detects even more correctly reported outliers than ACE. While ACHashACE can also report more correctly reported outliers than ACE, the number of outliers reported is much higher than the rest, indicating a lower precision. In addition, the detection latency of ACHashACE is inferior to FastACE.

To sum up, FastLSH can be applied to anomaly detection task. It significantly decreases the end-to-end detection latency and offers the same or even better performance than the state-of-the-art. More description and discussion about this set of experiments are presented in Appendix C.1.

Table 1: Results on Musk

| Methods | Outliers Reported | Correctly Reported | Outliers Missed | Execution Time (s) | Speedup |
|---|---|---|---|---|---|
| ACE | 310 | 73 | 14 | 0.2794 | 1x |
| ACHashACE | 292 | 51 | 46 | 0.1168 | 2.4x |
| FastACE | 304 | 74 | 13 | **0.04523** | **6.1**x |

Table 2: Results on a9a

| Methods | Outliers Reported | Correctly Reported | Outliers Missed | Execution Time (s) | Speed-up |
|---|---|---|---|---|---|
| ACE | 13850 | 4748 | 3093 | 2.079 | 1x |
| ACHashACE | 16877 | 5468 | 2373 | 0.9224 | 2.3x |
| FastACE | 15753 | 5469 | 2372 | **0.5138** | **4**x |

Table 3: Results on Statlog Shuttle

| Methods | Outliers Reported | Correctly Reported | Outliers Missed | Execution Time (s) | Speedup |
|---|---|---|---|---|---|
| ACE | 1931 | 472 | 407 | 0.3297 | 1x |
| ACHashACE | 2226 | 531 | 348 | 0.2924 | 1.1x |
| FastACE | 1822 | 520 | 359 | **0.2589** | **1.2**x |

**Neural Network Training:** For this task, we show the training time and the number of iterations for SLIDE, FastSLIDE (FastLSH + SLIDE) and ACHashSLIDE (ACHash + SILDE) in Figure 2. Note that the $x$-axis is in log-scale, and all the curves have a long flat converged portion when plotted on a linear scale indicating clear convergence behavior. We can see from the plots that FastSLIDE achieves around the same or even better classification accuracy compared with SLIDE, while enjoying 1.7x and 1.4x speedup over Delicious-200K and Amazon-670K in training, respectively. The performance gain comes from the efficiency of FastLSH – hash tables have to be constructed every $N_0$ iterations and FastLSH reduces such hashing cost significantly. In addition, although the performance of ACHashSLIDE is comparable or slightly better than SLIDE, it is inferior to FastSLIDE due to the lack of theoretical guarantee. Thus, FastLSH is well-suited for accelerating neural network training. More description and discussion about this set of experiments are given in Appendix C.2.

**Nearest Neighbor Search:** Next, we validate through the ANN search task why FastLSH reduces the end-to-end execution time for the two aforementioned tasks. The reason is that FastLSH can significantly reduce the end-to-end latency in index construction while achieving comparable query performance [2] to E2LSH. Hence, for many LSH-based applications, if the end-to-end execution time is more important than query time, the acceleration of FastLSH becomes particularly significant.

We first compare the performance among E2LSH, ACHash and FastLSH for 0.9 target recall [3]. The recall, average query time, LSH computation time and index construction time for *ImageNet*, *Trevi*, *Deep*, *Random*, *Glove* and *Ukbench* are illustrated in Figure 3 (a), (b), (c) and (d), respectively. As plotted in Figure 3 (a) and (b), FastLSH and E2LSH achieve comparable query performance and answer quality. Due to lack of theoretical guarantee, ACHash performs slightly worse than FastLSH and E2LSH in most cases w.r.t query efficiency.

---

[2]Note that FastLSH does not decrease the query time in the ANN search task because the cost in query processing is dominated by the exact evaluation of distances between candidates and the query.

[3]The actual recall may vary around 0.9 slightly

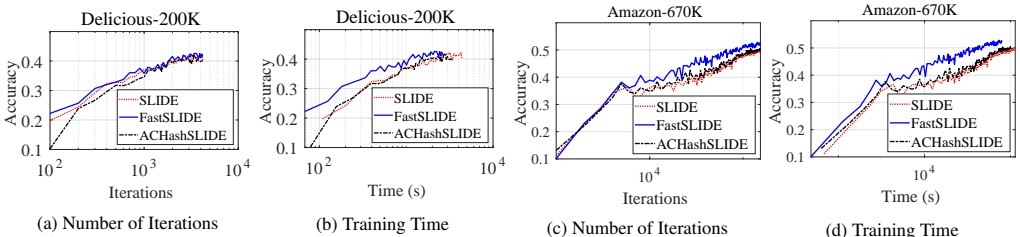

Figure 2: Comparison of SLIDE, FastSLIDE and ACHashSLIDE. The $x$-axis is plotted in log scale.

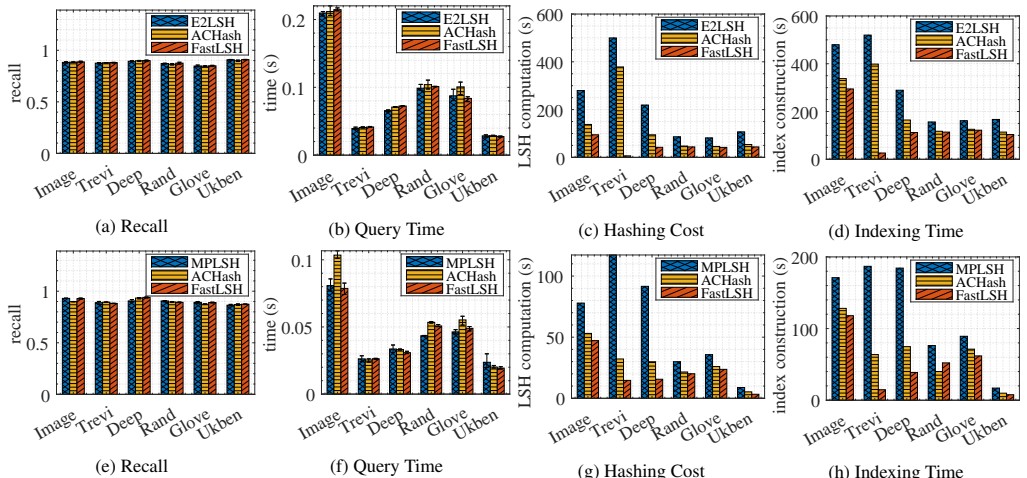

Figure 3: (a-d) Comparison with E2LSH and ACHash. (e-h) Comparison with MPLSH and ACHash.

The performance of the three methods differs dramatically when it turns to the cost of hashing and indexing time. As shown in Figure 3 (c), the LSH computation time of FastLSH is significantly superior to E2LSH and ACHash. For example, FastLSH obtains around 80 times speedup over E2LSH and runs 60 times faster than ACHash on *Trevi*. For ACHash, the fixed sampling ratio and overhead in Hadamard transform make it inferior to FastLSH. Similar trends can be found with MPLSH in Figure 3 (e), (f), (g) and (h).

The end-to-end speedup in the index construction time is illustrated in Figure 3 (d) and (h). Thanks to the significant drop in hashing cost, the time spent in building the index decreases by up to a factor of 20. Besides hashing, the procedure of index construction consists of other operations such as hash table initialization and linked list maintainance, which cannot be accelarated. Thus, the end-to-end latency in index construction decreases not as much as the hashing cost. More results on other datasets are deferred to Figure 4 in Appendix C.3 and Figure 6 in Appendix C.4.

We also plot the recall v.s. average query time curves by varying target recalls to obtain a complete picture of FastLSH. Please refer to Figure 5 in Appendix C.3 for more information. In addition, to make more comprehensive analysis of FastLSH, we explore its effectiveness in handling sparse data. The reasons and empirical evidence are presented in Appendix C.5.

## 7 CONCLUSION

In this paper, we develop FastLSH to accelerate hash function evaluation, which maintains the same theoretical guarantee and empirical performance as the classic E2LSH. Rigid analysis shows that the probability of collision of FastLSH is asymptotically equal to that of E2LSH. In the case of limited $m$, we quantitatively analyze the impact of $\sigma$ and $m$ on the probability of collision. Extensive experiments on a number of machine learning tasks demonstrate that FastLSH is a promising alternative to the classic LSH scheme.

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

APPENDIX

## A PROOFS

**Lemma A.1.** *The characteristic function of the product of two independent random variables $W = XY$ is as follows:*

$$\varphi_W(x) = E_Y\{\exp(-\frac{x^2 Y^2}{2})\}$$

*where $X$ is a standard normal random variable and $Y$ is an independent random variable with mean $\mu$ and variance $\sigma^2$.*

*Proof.* For the characteristic function of $W$, we can write:

$$
\begin{aligned}
\varphi_W(x) &= E_W\{\exp(ixW)\} \\
&= E_{XY}\{\exp(ixXY)\} \\
&= E_Y\{E_{X|Y}\{\exp(ixXY)|Y\}\} \\
&= \int_{-\infty}^{+\infty} E_{X|Y}\{\exp(-ixXY)|Y\} f(Y) dY \\
&= \int_{-\infty}^{+\infty} (\int_{-\infty}^{+\infty} \exp(-ixXY) f(X|Y) dX) f(Y) dY \\
&= \int_{-\infty}^{+\infty} \exp(-\frac{x^2 Y^2}{2}) f(Y) dY \\
&= E_Y\{\exp(-\frac{x^2 Y^2}{2})\}
\end{aligned}
$$

where standard normal random variable $X$ is eliminated by its characteristic function. We prove this Lemma. $\square$

**Lemma A.2.** *The characteristic function of $\tilde{s}X$ is as follows:*

$$\varphi_{\tilde{s}X}(x) = \frac{1}{2(1 - \Phi(\frac{-\tilde{\mu}}{\tilde{\sigma}}))} \exp(\frac{1}{8} x^4 \tilde{\sigma}^2 - \frac{1}{2}\tilde{\mu}x^2) \operatorname{erfc}(\frac{\frac{1}{2}x^2 \tilde{\sigma}^2 - \tilde{\mu}}{\sqrt{2}\tilde{\sigma}}) \quad (-\infty < x < +\infty)$$

*where $\operatorname{erfc}(t) = \frac{2}{\sqrt{\pi}} \int_t^{+\infty} \exp(-x^2) dx \ (-\infty < t < +\infty)$ is the complementary error function.*

*Proof.* According to Eqn. 8, we know the PDF of $\tilde{s}$. By applying Lemma A.1, we have the following result:

$$
\begin{aligned}
\varphi_{\tilde{s}X}(x) &= \frac{1}{\sqrt{2\pi}\tilde{\sigma}} \int_0^{+\infty} \frac{2y}{\Phi(a_2; \tilde{\mu}, \tilde{\sigma}^2) - \Phi(a_1; \tilde{\mu}, \tilde{\sigma}^2)} \exp(\frac{-x^2 y^2}{2} - \frac{(y^2 - \tilde{\mu})^2}{2\tilde{\sigma}^2}) dy \\
&= \frac{1}{\sqrt{2\pi}\tilde{\sigma}} \int_0^{+\infty} \frac{1}{\Phi(a_2; \tilde{\mu}, \tilde{\sigma}^2) - \Phi(a_1; \tilde{\mu}, \tilde{\sigma}^2)} \exp(-\frac{x^2 y^2}{2} - \frac{(y^2 - \tilde{\mu})^2}{2\tilde{\sigma}^2}) dy^2 \\
&= \frac{1}{\sqrt{2\pi}\tilde{\sigma}} \int_0^{+\infty} \frac{1}{\Phi(a_2; \tilde{\mu}, \tilde{\sigma}^2) - \Phi(a_1; \tilde{\mu}, \tilde{\sigma}^2)} \exp(\frac{-(y^2 - (\tilde{\mu} - \frac{1}{2}x^2\tilde{\sigma}^2))^2 - \tilde{\mu}x^2\tilde{\sigma}^2 + \frac{1}{4}x^4\tilde{\sigma}^4)}{2\tilde{\sigma}^2}) dy^2 \\
&= \frac{1}{\sqrt{2\pi}\tilde{\sigma}} \int_0^{+\infty} \frac{1}{\Phi(a_2; \tilde{\mu}, \tilde{\sigma}^2) - \Phi(a_1; \tilde{\mu}, \tilde{\sigma}^2)} \exp(\frac{1}{8}x^4\tilde{\sigma}^2 - \frac{1}{2}\tilde{\mu}x^2) \exp(\frac{-(y^2 - (\tilde{\mu} - \frac{1}{2}x^2\tilde{\sigma}^2))^2)}{2\tilde{\sigma}^2}) dy^2 \\
&= \frac{1}{2(\Phi(\frac{a_2 - \tilde{\mu}}{\tilde{\sigma}}) - \Phi(\frac{a_1 - \tilde{\mu}}{\tilde{\sigma}}))} \exp(\frac{1}{8}x^4\tilde{\sigma}^2 - \frac{1}{2}\tilde{\mu}x^2) \operatorname{erfc}(\frac{\frac{1}{2}x^2\tilde{\sigma}^2 - \tilde{\mu}}{\sqrt{2}\tilde{\sigma}}) \\
&= \frac{1}{2(1 - \Phi(\frac{-\tilde{\mu}}{\tilde{\sigma}}))} \exp(\frac{1}{8}x^4\tilde{\sigma}^2 - \frac{1}{2}\tilde{\mu}x^2) \operatorname{erfc}(\frac{\frac{1}{2}x^2\tilde{\sigma}^2 - \tilde{\mu}}{\sqrt{2}\tilde{\sigma}})
\end{aligned}
$$

where $\tilde{\mu} = \frac{ms^2}{n}$, $\tilde{\sigma}^2 = m\sigma^2$, $a_2 = \infty$ and $a_1 = 0$. Hence we prove this Lemma. $\square$

**Theorem A.3.** *The collision probability of FastLSH is as follows:*

$$p(s, \sigma) = Pr[h_{\tilde{a}, \tilde{b}}(\boldsymbol{v}) = h_{\tilde{a}, \tilde{b}}(\boldsymbol{u})] = \int_0^{\tilde{w}} f_{|\tilde{s}X|}(t)(1 - \frac{t}{\tilde{w}})dt$$

*Proof.* Let $f_{|\tilde{s}X|}(t)$ represent the PDF of the absolute value of $\tilde{s}X$. For given bucket width $\tilde{w}$, the probability $p(|\tilde{s}X| < t)$ for any pair $(\boldsymbol{v}, \boldsymbol{u})$ is computed as $p(|\tilde{s}X| < t) = \int_0^{\tilde{w}} f_{|\tilde{s}X|}(t)dt$, where $t \in [0, \tilde{w}]$. Recall that $\tilde{b}$ follows the uniform distribution $U(0, \tilde{w})$, the probability $p(\tilde{b} < \tilde{w} - t)$ is thus $(1 - \frac{t}{\tilde{w}})$. This means that after random projection, $(|\tilde{s}X| + \tilde{b})$ is also within the same bucket, and the collision probability $p(s, \sigma)$ is the product of $p(|\tilde{s}X| < t)$ and $p(\tilde{b} < \tilde{w} - t)$. Hence we prove this Theorem. □

**Theorem A.4.**

$$\lim_{m \to +\infty} \frac{\varphi_{\tilde{s}X}(x)}{\exp(-\frac{ms^2x^2}{2n})} = 1$$

*where $\exp(-\frac{ms^2x^2}{2n})$ is the characteristic function of $\mathcal{N}(0, \frac{ms^2}{n})$.*

*Proof.* Recall that $\tilde{\mu} = \frac{ms^2}{n}$ and $\tilde{\sigma} = m\sigma^2$. $\varphi_{\tilde{s}X}(x)$ can be written as follows:

$$\varphi_{\tilde{s}X}(x) = \frac{1}{2(1 - \Phi(\frac{-\sqrt{ms^2}}{n\sigma}))} \exp(\frac{mx^4\sigma^2}{8} - \frac{ms^2x^2}{2n}) \operatorname{erfc}(\frac{\sqrt{m}(nx^2\sigma^2 - 2s^2)}{2\sqrt{2}n\sigma}) \quad (-\infty < x < +\infty)$$

Let $a = \frac{s^2}{\sqrt{2}n\sigma} > 0$ and $b = \frac{\sigma}{2\sqrt{2}} > 0 \Rightarrow b^2 = \frac{\sigma^2}{8}$. According to the fact $\Phi(x) = \frac{1}{2}\operatorname{erfc}(-\frac{x}{\sqrt{2}})$, $\varphi_{\tilde{s}X}(x)$ is simplified as:

$$\varphi_{\tilde{s}X}(x) = \exp(b^2mx^4 - \frac{1}{2}m\mu x^2) \cdot \frac{\operatorname{erfc}(b\sqrt{m}x^2 - a\sqrt{m})}{2 - \operatorname{erfc}(a\sqrt{m})}$$

Let $g(x) = \frac{\varphi_{\tilde{s}X}(x)}{\varphi_{sX}(x)}$, where $\varphi_{sX}(x) = \exp(-\frac{ms^2x^2}{2n})$ is the characteristic function of $\mathcal{N}(0, \frac{ms^2}{n})$. Then $g(x)$ is denoted as:

$$g(x) = \exp(b^2mx^4) \cdot \frac{\operatorname{erfc}(b\sqrt{m}x^2 - a\sqrt{m})}{2 - \operatorname{erfc}(a\sqrt{m})}$$

To prove $\varphi_{\tilde{s}X}(x) = \varphi_{sX}(x)$, we convert to prove whether $g(x) = 1$ as $m \to \infty$. Obviously $x^2 \leq O(m^{-1})$. Then $\sqrt{m}x^2 \leq O(m^{-1/2})$ and $mx^4 \leq O(m^{-1})$. It is easy to derive:

$$\lim_{m \to +\infty} \exp(b^2mx^4) = 1$$

It holds for any fixed $b \in \mathbb{R}^+$. On the other hand, we have:

$$\operatorname{erfc}(b\sqrt{m}x^2 - a\sqrt{m}) \sim \operatorname{erfc}(-a\sqrt{m}), \quad m \to +\infty$$

Actually using the fact $b\sqrt{m}x^2 - a\sqrt{m} \sim -a\sqrt{m}$ as $m \to +\infty$ and $\operatorname{erfc}(-x) = 2 - \frac{\exp(-x^2)}{\sqrt{\pi}x}$ as $x \to +\infty$, we have:

$$\lim_{m \to +\infty} \frac{\operatorname{erfc}(b\sqrt{m}x^2 - a\sqrt{m})}{2 - \operatorname{erfc}(a\sqrt{m})} = \frac{\lim_{m \to +\infty} \operatorname{erfc}(b\sqrt{m}x^2 - a\sqrt{m})}{\lim_{m \to +\infty} \operatorname{erfc}(-a\sqrt{m})} = 1$$

Hence we have:

$$g(x) = \lim_{m \to +\infty} \exp(b^2mx^4) \cdot \lim_{m \to +\infty} \frac{\operatorname{erfc}(b\sqrt{m}x^2 - a\sqrt{m})}{\operatorname{erfc}(-a\sqrt{m})} = 1.$$

We prove this Theorem. □

If the characteristic function of a random variable $Z$ exists, it provides a way to compute its various moments. Specifically, the $r$-th moment of $Z$ denoted by $E(Z^r)$ can be expressed as the $r$-th derivative of the characteristic function evaluated at zero Hoel et al. (1971), i.e.,

$$E(Z^r) = (i)^{-r} \frac{d^r}{dt^r} \varphi_Z(t) \mid_{t=0} \tag{10}$$

where $\varphi_Z(t)$ denotes the characteristic function of $Z$. If we know the characteristic function, then all of the moments of the random variable $Z$ can be obtained.

**Lemma A.5.**

$$\begin{cases} E(\tilde{s}X) &= 0 \\ E((\tilde{s}X)^2) = \frac{ms^2}{n}(1+\epsilon) \\ E((\tilde{s}X)^3) = 0 \\ E((\tilde{s}X)^4) = \frac{3m^2 s^4}{n^2}(1+\lambda) \end{cases}$$

where $\epsilon = \frac{\tilde{\sigma} \exp(\frac{-\tilde{\mu}^2}{2\tilde{\sigma}^2})}{\sqrt{2\pi}\tilde{\mu}(1-\Phi(\frac{-\tilde{\mu}}{\tilde{\sigma}}))}$ and $\lambda = \frac{\tilde{\sigma}^2}{\tilde{\mu}^2} + \epsilon$.

*Proof.* From Lemma A.2, we can easily compute the first-order derivative of characteristic function with respect to $x$, which is as follows:

$$\varphi'_{\tilde{s}X}(x) = \frac{\exp(\frac{1}{8}x^4\tilde{\sigma}^2 - \frac{1}{2}\tilde{\mu}x^2)}{2(1-\Phi(\frac{-\tilde{\mu}}{\tilde{\sigma}}))} \left[ (\frac{1}{2}x^3\tilde{\sigma}^2 - \tilde{\mu}x) \operatorname{erfc}(\frac{\frac{1}{2}x^2\tilde{\sigma}^2 - \tilde{\mu}}{\sqrt{2}\tilde{\sigma}}) - \frac{2\tilde{\sigma}x}{\sqrt{2\pi}} \exp(-(\frac{\frac{1}{2}x^2\tilde{\sigma}^2 - \tilde{\mu}}{\sqrt{2}\tilde{\sigma}})^2)) \right] \tag{11}$$

Then the second-order derivative is

$$\varphi''_{\tilde{s}X}(x) = \frac{\exp(\frac{1}{8}x^4\tilde{\sigma}^2 - \frac{1}{2}\tilde{\mu}x^2)}{2(1-\Phi(\frac{-\tilde{\mu}}{\tilde{\sigma}}))} \left[ ((\frac{1}{2}x^3\tilde{\sigma}^2 - ux)^2 + \frac{3}{2}x^2\tilde{\sigma}^2 - \tilde{\mu}) \operatorname{erfc}(\frac{\frac{1}{2}x^2\tilde{\sigma}^2 - \tilde{\mu}}{\sqrt{2}\tilde{\sigma}}) \right.$$
$$\left. - \frac{2}{\sqrt{2\pi}}(\tilde{\sigma} + \frac{3}{2}x^4\tilde{\sigma}^3 - \frac{3}{2}\tilde{\mu}\tilde{\sigma}x^2) \exp(-(\frac{\frac{1}{2}x^2\tilde{\sigma}^2 - \tilde{\mu}}{\sqrt{2}\tilde{\sigma}})^2)) \right] \tag{12}$$

The third-order derivative is

$$\varphi'''(x) = \frac{\exp(\frac{1}{8}x^4\tilde{\sigma}^2 - \frac{1}{2}\tilde{\mu}x^2)}{2(1-\Phi(\frac{-\tilde{\mu}}{\tilde{\sigma}}))} \left[ (\frac{1}{8}\tilde{\sigma}^6 x^9 - \frac{3}{4}\tilde{\mu}\tilde{\sigma}^4 x^7 + (\frac{2}{3}\tilde{\mu}^2\tilde{\sigma}^2 + \frac{9}{4}\tilde{\sigma}^4)x^5 - (6\tilde{\mu}\tilde{\sigma}^2 + \tilde{\mu}^3)x^3 + 3(\tilde{\mu}^2 + \tilde{\sigma}^2)x) \right.$$
$$\left. \operatorname{erfc}(\frac{\frac{1}{2}\tilde{\sigma}^2 x^2 - \tilde{\mu}}{\sqrt{2}\tilde{\sigma}}) - \frac{2}{\sqrt{2\pi}}(\frac{1}{4}\tilde{\sigma}^5 x^7 - \tilde{\mu}\tilde{\sigma}^3 x^5 + (\tilde{\mu}^2\tilde{\sigma} + \frac{7}{2}\tilde{\sigma}^3)x^3 - 3\tilde{\mu}\tilde{\sigma}x) \exp(-(\frac{\frac{1}{2}\tilde{\sigma}^2 x^2 - \tilde{\mu}}{\sqrt{2}\tilde{\sigma}})^2)) \right] \tag{13}$$

The fourth-order derivative is

$$\varphi''''(x) = \frac{\exp(\frac{1}{8}x^4\tilde{\sigma}^2 - \frac{1}{2}\tilde{\mu}x^2)}{2(1-\Phi(\frac{-\tilde{\mu}}{\tilde{\sigma}}))} \left[ (\frac{1}{16}\tilde{\sigma}^8 x^{12} - \frac{1}{2}\tilde{\mu}\tilde{\sigma}^6 x^{10} + (\frac{3}{2}\tilde{\mu}^2\tilde{\sigma}^4 + \frac{9}{4}\tilde{\sigma}^4)x^8 - (2\tilde{\mu}^3\tilde{\sigma}^2 + \frac{21}{2}\tilde{\mu}\tilde{\sigma}^4)x^6 \right.$$
$$+ (\tilde{\mu}^4 + \frac{51}{4}\tilde{\sigma}^4 + 9\tilde{\mu}^2\tilde{\sigma}^2)x^4 - (6\tilde{\mu}^3 + 21\tilde{\mu}\tilde{\sigma}^2)x^2 + 3(\tilde{\mu}^2 + \tilde{\sigma}^2)) \operatorname{erfc}(\frac{\frac{1}{2}\tilde{\sigma}^2 x^2 - \tilde{\mu}}{\sqrt{2}\tilde{\sigma}})$$
$$\left. - \frac{2}{\sqrt{2\pi}}(\frac{1}{8}\tilde{\sigma}^7 x^{10} - \frac{3}{4}\tilde{\mu}\tilde{\sigma}^5 x^8 + (\frac{3}{2}\tilde{\mu}^2\tilde{\sigma}^3 + 4\tilde{\sigma}^5)x^6 - (6\tilde{\mu}^2\tilde{\sigma} + \frac{13}{2}\tilde{\sigma}^3)x^2 - 3\tilde{\mu}\tilde{\sigma}) \exp(-(\frac{\frac{1}{2}\tilde{\sigma}^2 x^2 - \tilde{\mu}}{\sqrt{2}\tilde{\sigma}})^2)) \right] \tag{14}$$

Let $E(\tilde{s}X - E(\tilde{s}X))^i$ for $i \in \{1, 2, 3, 4\}$ denote the first four central moments. According to Eqn. 10, we know that $E(\tilde{s}X) = \frac{\varphi'(0)}{i} = 0$, then it is easy to derive $E(\tilde{s}X - E(\tilde{s}X))^i = E((\tilde{s}X)^i)$. To this end, by Eqn. 11 - 14, we have the following results:

$$\begin{cases} E(\tilde{s}X) &= 0 \\ E((\tilde{s}X)^2) = \frac{\tilde{\mu}\operatorname{erfc}(\frac{-\tilde{\mu}}{\sqrt{2}\tilde{\sigma}}) + \frac{2\tilde{\sigma}}{\sqrt{2\pi}}\exp(\frac{-\tilde{\mu}^2}{2\tilde{\sigma}^2})}{2(1-\Phi(\frac{-\tilde{\mu}}{\tilde{\sigma}}))} \\ E((\tilde{s}X)^3) = 0 \\ E((\tilde{s}X)^4) = \frac{3(\tilde{\mu}^2 + \tilde{\sigma}^2)\operatorname{erfc}(\frac{-\tilde{\mu}}{\sqrt{2}\tilde{\sigma}}) + \frac{6\tilde{\mu}\tilde{\sigma}}{\sqrt{2\pi}}\exp(\frac{-\tilde{\mu}^2}{2\tilde{\sigma}^2})}{2(1-\Phi(\frac{-\tilde{\mu}}{\tilde{\sigma}}))} \end{cases} \tag{15}$$

where $E(\tilde{s}X)$ is the expectation; $E((\tilde{s}X)^2)$ is the variance; $\frac{E((\tilde{s}X)^3)}{(E((\tilde{s}X)^2))^{\frac{3}{2}}}$ is the skewness; $\frac{E((\tilde{s}X)^4)}{(E((\tilde{s}X)^2))^2}$ is the kurtosis. Since

$$\frac{\text{erfc}(\frac{-\tilde{\mu}}{\sqrt{2}\tilde{\sigma}})}{2(1-\Phi(\frac{-\tilde{\mu}}{\tilde{\sigma}}))} = \frac{\frac{2}{\sqrt{\pi}}\int_{\frac{-\tilde{\mu}}{\sqrt{2}\tilde{\sigma}}}^{+\infty}\exp(-t_1^2)dt_1}{2(1-\frac{1}{\sqrt{2\pi}}\int_{-\infty}^{\frac{-\tilde{\mu}}{\tilde{\sigma}}}\exp(\frac{-t_2^2}{2})dt_2)} = \frac{\int_{\frac{-\tilde{\mu}}{\sqrt{2}\tilde{\sigma}}}^{+\infty}\exp(-t_1^2)dt_1}{\sqrt{2}\int_{\frac{-\tilde{\mu}}{\tilde{\sigma}}}^{+\infty}\exp(\frac{-t_2^2}{2})dt_2} = \frac{\int_{\frac{-\tilde{\mu}}{\sqrt{2}\tilde{\sigma}}}^{+\infty}\exp(-t_1^2)dt_1}{\int_{\frac{-\tilde{\mu}}{\sqrt{2}\tilde{\sigma}}}^{+\infty}\exp(-t^2)dt} = 1$$

Therefore, Eqn. 15 can be rewritten as below

$$\begin{cases} E(\tilde{s}X) & = 0 \\ E((\tilde{s}X)^2) = \tilde{\mu}(1+\epsilon) \\ E((\tilde{s}X)^3) = 0 \\ E((\tilde{s}X)^4) = 3\tilde{\mu}^2(1+\lambda) \end{cases}$$

where $\epsilon = \frac{\tilde{\sigma}\exp(\frac{-\tilde{\mu}^2}{2\tilde{\sigma}^2})}{\sqrt{2\pi}\tilde{\mu}(1-\Phi(\frac{-\tilde{\mu}}{\tilde{\sigma}}))}$, $\lambda = \frac{\tilde{\sigma}^2}{\tilde{\mu}^2} + \epsilon$, $\tilde{\mu} = \frac{ms^2}{n}$ and $\tilde{\sigma}^2 = m\sigma^2$. We prove this Lemma.

$\square$

# B EXTENSION OF FASTLSH

## B.1 EXTENSION TO MAXIMUM INNER PRODUCT

Bachrach et al. (2014); Shrivastava & Li (2014) shows that there exists two transformation functions, by which the maximum inner product search (MIPS) problem can be converted into solve the near neighbor search problem. More specifically, the two transformation functions are $P(\boldsymbol{v}) = (\sqrt{\kappa^2 - \|\boldsymbol{v}\|_2^2}, \boldsymbol{v})$ for data processing and $Q(\boldsymbol{u}) = (0, \boldsymbol{u})$ for query processing respectively, where $\kappa = max(\|\boldsymbol{v}_i\|_2)$ ($i \in \{1, 2, \ldots, N\}$). Then the relationship between maximum inner product and $l_2$ norm for any vector pair $(\boldsymbol{v}_i, \boldsymbol{u})$ is denoted as $\underset{i}{argmax}(\frac{P(\boldsymbol{v}_i)Q(\boldsymbol{u})}{\|P(\boldsymbol{v}_i)\|_2\|Q(\boldsymbol{u})\|_2}) = \underset{i}{argmin}(\|P(\boldsymbol{v}_i) - Q(\boldsymbol{u})\|_2)$ for $\|\boldsymbol{u}\|_2 = 1$. To make FastLSH applicable for MIPS, we first apply the sample operator $S(\cdot)$ defined earlier to vector pairs $(\boldsymbol{v}_i, \boldsymbol{u})$ for yielding $\tilde{\boldsymbol{v}}_i = S(\boldsymbol{v}_i)$ and $\tilde{\boldsymbol{u}} = S(\boldsymbol{u})$, and then obtain $\tilde{P}(\boldsymbol{v}) = (\sqrt{\tilde{\kappa}^2 - \|S(\boldsymbol{v})\|_2^2}, S(\boldsymbol{v}))$ and $\tilde{Q}(\boldsymbol{u}) = (0, S(\boldsymbol{u}))$, where $\tilde{\kappa} = max(\|S(\boldsymbol{v}_i)\|_2)$ is a constant. Then $\underset{i}{argmax}(\frac{\tilde{P}(\boldsymbol{v}_i)\tilde{Q}(\boldsymbol{u})}{\|\tilde{P}(\boldsymbol{v}_i)\|_2\|\tilde{Q}(\boldsymbol{u})\|_2}) = \underset{i}{argmin}(\|\tilde{P}(\boldsymbol{v}_i) - \tilde{Q}(\boldsymbol{u})\|_2)$ for $\|S(\boldsymbol{u})\|_2 = 1$. Let $\triangle = \tilde{\kappa}^2 - \|S(\boldsymbol{v})\|_2^2$. After random projection, $\boldsymbol{a}^T\tilde{P}(\boldsymbol{v}) - \boldsymbol{a}^T\tilde{Q}(\boldsymbol{u})$ is distributed as $(\sqrt{\tilde{s}^2 + \triangle})X$. Since $\tilde{s}^2 \sim \mathcal{N}(m\mu, m\sigma^2)$, then $(\tilde{s}^2 + \triangle) \sim \mathcal{N}(m\mu + \triangle, m\sigma^2)$. Let $\sqrt{\tilde{s}^2 + \triangle}$ be the random variable $\mathcal{I}$. Similar to Eqn. 8, the PDF of $\mathcal{I}$ represented by $f_{\mathcal{I}}$ is yielded as follow:

$$f_{\mathcal{I}}(t) = 2t\psi(t^2; m\mu + \triangle, m\sigma^2, 0, +\infty) \tag{16}$$

By applying Lemma A.1, the characteristic function of $\mathcal{I}X$ is as follows:

$$\varphi_{\mathcal{I}X}(x) = \frac{1}{2(1-\Phi(\frac{-ms^2-n\triangle}{\sqrt{m}\sigma\triangle}))}\exp(\frac{mx^4\sigma^2}{8} - \frac{(ms^2+n\triangle)x^2}{2n})\text{erfc}(\frac{mnx^2\sigma^2 - 2(ms^2+n\triangle)}{2\sqrt{2mn}\sigma})(-\infty < x < +\infty) \tag{17}$$

Then the PDF of $\mathcal{I}X$ is obtained by $\varphi_{\mathcal{I}X}(x)$:

$$f_{\mathcal{I}X}(t) = \frac{1}{2\pi}\int_{-\infty}^{\infty}\varphi_{\mathcal{I}X}(x)exp(-itx)dx \tag{18}$$

It is easy to derive the collision probability of any pair $(\boldsymbol{v}, \boldsymbol{u})$ by $f_{\mathcal{I}X}(t)$, which is as follows:

$$p(s) = \int_{0}^{\tilde{w}'}f_{|\mathcal{I}X|}(t)(1 - \frac{t}{\tilde{w}'})dx \tag{19}$$

where $f_{|\mathcal{I}X|}(t)$ denotes the PDF of the absolute value of $\mathcal{I}X$. $\tilde{w}'$ is the bucket width.

## B.2 EXTENSION TO $l_p$ NORM FOR $p \in (0, 2)$

The analysis of FastLSH extended to $l_p$ norm ($p \in (0, 2)$) is similar to that of $l_2$ norm, it is given as follows:

For given vector pair $(\boldsymbol{v}, \boldsymbol{u})$, let $s = \|\boldsymbol{v} - \boldsymbol{u}\|_p$, where $p \in (0, 2)$. The collection of $n$ entries $(v_i - u_i)^p$ $\{i = 1, 2, \ldots, n\}$ follows an unknown distribution with a finite mean $\mu_p = (\sum_{i=1}^n (v_i - u_i)^p)/n$ and variance $\sigma_p^2 = (\sum_{i=1}^n ((v_i - u_i)^p - \mu_p)^2)/n$. After performing the sampling operator $S(\cdot)$ of size $m$, $\boldsymbol{v}$ and $\boldsymbol{u}$ are transformed into $\tilde{\boldsymbol{v}} = S(\boldsymbol{v})$ and $\tilde{\boldsymbol{u}} = S(\boldsymbol{u})$, and the $l_p$ distance of $(\tilde{\boldsymbol{v}}, \tilde{\boldsymbol{u}})$ is $\tilde{s}^p = \sum_{i=1}^m (\tilde{v}_i - \tilde{u}_i)^p$. By Central Limit Theorem, we have the following lemma:

**Lemma B.1.** *If $m$ is sufficiently large, then the sum $\tilde{s}^p$ of $m$ i.i.d. random samples $(\tilde{v}_i - \tilde{u}_i)^p$ ($i \in 1, 2, \ldots, m$) converges asymptotically to the normal distribution with mean $m\mu_p$ and variance $m\sigma_p^2$, i.e., $\tilde{s}^p \sim \mathcal{N}(m\mu_p, m\sigma_p^2)$, where $p \in (0, 2)$.*

Since $\tilde{s}^p \geq 0$, $\tilde{s}^p$ can be modeled by normal distribution $\tilde{s}^p \sim \mathcal{N}(m\mu_p, m\sigma_p^2)$ over the truncation interval $[0, +\infty)$, that is, the singly-truncated normal distribution $\psi(x; \tilde{\mu}_p, \tilde{\sigma}_p^2, 0, +\infty)$. Considering the fact that $\tilde{s} \geq 0$, we have $Pr[\tilde{s} < t] = Pr[\tilde{s}^p < t^p]$ for any $t > 0$. Therefore, the CDF of $\tilde{s}$ for $p \in (0, 2)$, denoted by $F_{\tilde{s}}^p$, can be computed as follows:

$$\begin{aligned} F_{\tilde{s}}^p(t) = Pr[\tilde{s} < t] &= Pr[\tilde{s}^p < t^p] \\ &= \int_0^{t^p} \psi(x; \tilde{\mu}_p, \tilde{\sigma}_p^2, 0, \infty) dx \end{aligned} \tag{20}$$

where $\tilde{\mu}_p = m\mu_p$ and $\tilde{\sigma}_p^2 = m\sigma_p^2$. Due to the fact that the PDF is the derivative of the CDF, the PDF of $\tilde{s}$ for $p \in (0, 2)$, denoted by $f_{\tilde{s}}^p$, is derived as follows:

$$f_{\tilde{s}}^p(t) = \frac{d}{dt}[F_{\tilde{s}}^p(t)] = pt^{p-1}\psi(t^p; \tilde{\mu}_p, \tilde{\sigma}_p^2, 0, \infty) \tag{21}$$

If $\boldsymbol{a}$ is a projection vector with entries drawn from a $p$-stable distribution, the distance between projections, $(\boldsymbol{a}^T \boldsymbol{v} - \boldsymbol{a}^T \boldsymbol{u})$, for two vectors $\boldsymbol{v}$ and $\boldsymbol{u}$, is distributed as $\|\boldsymbol{v} - \boldsymbol{u}\|_p X$, i.e., $sX$, where $X$ follows a $p$-stable distribution for $p \in (0, 2)$. Similarly, the projection distance between $\tilde{\boldsymbol{v}}$ and $\tilde{\boldsymbol{u}}$ under the projection vector $\tilde{\boldsymbol{a}}$ is given by $(\tilde{\boldsymbol{a}}^T \tilde{\boldsymbol{v}} - \tilde{\boldsymbol{a}}^T \tilde{\boldsymbol{u}})$, which follows the distribution $\tilde{s}X$. Thus, if we know the PDF of $\tilde{s}X$, we can easily derive the collision probability under the $l_p$ norm for the vector pair $(\boldsymbol{v}, \boldsymbol{u})$. However, for $p$-stable distributions where $p$ is not equal to $1/2$, $1$, or $2$, there is no closed-form expression for the PDF, making it intractable to derive the collision probability analytically for both classic LSH (E2LSH) and FastLSH Datar et al. (2004). Next, we focus on $p = 1/2$ and $p = 1$ to determine the distribution of $\tilde{s}X$ and analyze the asymptotic behavior of E2LSH and FastLSH, thereby indicates that FastLSH is asymptotically equivalent with E2LSH under the $l_p$ norm for $p \in (0, 2)$.

### B.2.1 $l_1$ NORM

We first need to use the following Lemma B.2 to obtain the characteristic function of $\tilde{s}X$ under $l_1$ norm, as the distribution of a random variable is determined uniquely by its characteristic function.

**Lemma B.2.** *The characteristic function of the product of two independent random variables $W = XY$ is as follows:*

$$\varphi_W(x) = E_Y\{exp(-|xY|)\}$$

*where $X$ is a standard Cauchy random variable and $Y$ is an independent random variable with mean $\mu$ and variance $\sigma^2$.*

*Proof.* This proof is similar to Lemma A.1. □

By Lemma B.2, we derive the characteristic function of $\tilde{s}X$, as shown in Lemma B.3. In the following derivation, we will slightly abuse the notation by using $\tilde{\mu}$ and $\tilde{\sigma}^2$, i.e., $\tilde{\mu} = \tilde{\mu}_1$ and $\tilde{\sigma}^2 = \tilde{\sigma}_1^2$.

**Lemma B.3.** *The characteristic function of $\tilde{s}X$ under $l_1$ norm is as follows:*

$$\varphi_{\tilde{s}X}(x) = \frac{1}{2(1 - \Phi(\frac{-\tilde{\mu}}{\tilde{\sigma}}))} exp(\frac{-\tilde{\mu}^2 + (\tilde{\mu} - \tilde{\sigma}^2|x|)^2}{2\tilde{\sigma}^2}) \operatorname{erfc}(\frac{\tilde{\sigma}^2|x| - \tilde{\mu}}{\sqrt{2}\tilde{\sigma}}) \quad (-\infty < x < +\infty)$$

*where $\tilde{\mu} = \frac{ms}{n}$ and $\tilde{\sigma}^2 = m\sigma_1^2$; $\operatorname{erfc}(t) = \frac{2}{\sqrt{\pi}} \int_t^{+\infty} \exp(-x^2)dx \ (-\infty < t < +\infty)$ is the complementary error function.*

*Proof.* According to Eqn. 21, we know the PDF of $\tilde{s}$ under $l_1$ norm. By applying Lemma B.3, we have:

$$\varphi_{\tilde{s}X}(x) = \frac{1}{\sqrt{2\pi}\tilde{\sigma}} \int_0^{+\infty} \frac{\exp(-|x|y - \frac{(y-\tilde{\mu})^2}{2\tilde{\sigma}^2})}{\Phi(a_2; \tilde{\mu}, \tilde{\sigma}^2) - \Phi(a_1; \tilde{\mu}, \tilde{\sigma}^2)} dy$$

$$= \frac{1}{\sqrt{2\pi}\tilde{\sigma}} \int_0^{+\infty} \frac{\exp(-\frac{(y^2 - 2\tilde{\mu}y + 2|x|\tilde{\sigma}^2 y + \tilde{\mu}^2)}{2\tilde{\sigma}^2})}{\Phi(a_2; \tilde{\mu}, \tilde{\sigma}^2) - \Phi(a_1; \tilde{\mu}, \tilde{\sigma}^2)} dy$$

$$= \frac{1}{\sqrt{2\pi}\tilde{\sigma}} \int_0^{+\infty} \frac{\exp(\frac{-\tilde{\mu} + (\tilde{\mu} - |x|\tilde{\sigma}^2)^2}{2\sigma^2}) \cdot \exp(-\frac{(y - (\tilde{\mu} - |x|\tilde{\sigma}^2))^2}{2\tilde{\sigma}^2})}{\Phi(a_2; \tilde{\mu}, \tilde{\sigma}^2) - \Phi(a_1; \tilde{\mu}, \tilde{\sigma}^2)} dy$$

$$= \frac{1}{2(1 - \Phi(\frac{-\tilde{\mu}}{\tilde{\sigma}}))} \cdot \exp(\frac{-\tilde{\mu} + (\tilde{\mu} - |x|\tilde{\sigma}^2)^2}{2\sigma^2}) \cdot \operatorname{erfc}(\frac{|x|\tilde{\sigma}^2 - \tilde{\mu}}{\sqrt{2}\tilde{\sigma}})$$

where $\tilde{\mu} = \frac{ms}{n}$, $\tilde{\sigma}^2 = m\sigma_1^2$, $a_2 = \infty$ and $a_1 = 0$. Hence we prove this Lemma $\qquad \square$

Given the characteristic function $\varphi_{\tilde{s}X}(x)$ under the $l_1$ norm, the probability density function (denoted by $c_{\tilde{s}X}(t)$) of $\tilde{s}X$ can be obtained by Eqn. 9. As a result, the collision probability for a pair $(\boldsymbol{v}, \boldsymbol{u})$ is computed as that of Theorem A.3, in which $f_{|\tilde{s}X|}(t)$ is replaced with $c_{|\tilde{s}X|}(t)$.

Similar to Theorem A.4, the following theorem gives the asymptotic behavior of the characteristic function of $\tilde{s}X$ under the $l_1$ norm, that is:

**Theorem B.4.**

$$\lim_{m \to +\infty} \frac{\varphi_{\tilde{s}X}(x)}{\exp(-\frac{ms|x|}{n})} = 1$$

*where $\exp(-\frac{ms|x|}{n})$ is the characteristic function of Cauchy distribution $\mathcal{C}(0, \frac{ms}{n})$.*

*Proof.* Given that $\tilde{\mu} = \tilde{\mu}_1 = \frac{ms}{n}$ and $\tilde{\sigma}^2 = m\sigma_1^2$. Let $\sigma^2 = \sigma_1^2$. According to Lemma B.3, we can express $\varphi_{\tilde{s}X}(x)$ in the following form:

$$\varphi_{\tilde{s}X}(x) = \frac{1}{2(1 - \Phi(\frac{-\sqrt{m}s}{n\sigma}))} \exp(\frac{1}{2}m\sigma^2 x^2 - \frac{ms|x|}{n}) \operatorname{erfc}(\frac{m\sigma^2|x| - ms/n}{\sqrt{2m}\sigma})$$

Let $a = \frac{s}{\sqrt{2}n\sigma} > 0$ and $b = \frac{\sigma}{\sqrt{2}} > 0 \Rightarrow b^2 = \frac{1}{2}\sigma^2$. Using the known fact that $\Phi(x) = \frac{1}{2}\operatorname{erfc}\left(\frac{-x}{\sqrt{2}}\right)$, we can simplify $\varphi_{\tilde{s}X}(x)$ as follows:

$$\varphi_{\tilde{s}X}(x) = \exp(b^2 m x^2 - \frac{ms|x|}{n}) \cdot \frac{\operatorname{erfc}(b\sqrt{m}|x| - a\sqrt{m})}{2 - \operatorname{erfc}(a\sqrt{m})}$$

Next, we define $g(x) = \frac{\varphi_{\tilde{s}X}(x)}{\varphi_{sX}(x)}$, where $\varphi_{sX}(x) = \exp(-\frac{ms|x|}{n})$ represents the characteristic function of the Cauchy distribution $\mathcal{C}(0, \frac{ms}{n})$. Therefore, $g(x)$ can be written as:

$$g(x) = \exp(b^2 m x^2) \cdot \frac{\operatorname{erfc}(b\sqrt{m}|x| - a\sqrt{m})}{2 - \operatorname{erfc}(a\sqrt{m})}$$

To prove $\varphi_{\tilde{s}X}(x) = \varphi_{sX}(x)$, we aim to show that as $m \to +\infty$, $g(x) \to 1$. Obviously $x \leq O(m^{-1})$, we have $\sqrt{m}|x| \leq O(m^{-1/2})$ and $mx^2 \leq O(m^{-1})$. Thus, for any constant $b \in \mathbb{R}^+$, the following holds:

$$\lim_{m \to +\infty} \exp(b^2 m x^2) = 1$$

On the other hand, we have:

$$\text{erfc}(b\sqrt{m}|x| - a\sqrt{m}) \sim \text{erfc}(-a\sqrt{m}) \qquad m \to +\infty$$

Since $b\sqrt{m}|x| - a\sqrt{m} \sim -a\sqrt{m}$ as $m \to +\infty$, and knowing that $\text{erfc}(-x) \sim 2 - \frac{\exp(-x^2)}{\sqrt{\pi}x}$ as $x \to +\infty$, we obtain:

$$\lim_{m \to +\infty} \frac{\text{erfc}(b\sqrt{m}|x| - a\sqrt{m})}{2 - \text{erfc}(a\sqrt{m})} = \frac{\lim_{m \to +\infty} \text{erfc}(b\sqrt{m}|x| - a\sqrt{m})}{\lim_{m \to +\infty} \text{erfc}(-a\sqrt{m})} = 1$$

Then, we have:

$$g(x) = \lim_{m \to +\infty} \exp(b^2 m x^2) \cdot \lim_{m \to +\infty} \frac{\text{erfc}(b\sqrt{m}|x| - a\sqrt{m})}{\text{erfc}(-a\sqrt{m})} = 1$$

Hence, we prove this Theorem. $\qquad\square$

Theorem B.4 means that $c_{\tilde{s}X}(t)$ is asymptotically equivalent to $\mathcal{C}(0, \frac{ms}{n})$. Then the following Corollary is naturally yielded:

**Corollary B.5.** $c_{\tilde{s}X}(t) \sim$ *the PDF of* $\mathcal{C}(0, \frac{ms}{n})$ *as $m$ tends to infinity.*

### B.2.2 $l_{1/2}$ NORM

Similar to that of $l_1$ norm, we use the following Lemma B.6 to derive the characteristic function of $\tilde{s}X$ under $l_{1/2}$ norm, that is:

**Lemma B.6.** *The characteristic function of the product of two independent random variables $W = XY$ is as follows:*

$$\varphi_W(x) = E_Y\{\exp(-\sqrt{2ixY})\}$$

*where $X$ is a standard Lévy random variable and $Y$ is an independent random variable with mean $\mu$ and variance $\sigma^2$, and the symbol $i = \sqrt{-1}$ represents the imaginary unit.*

*Proof.* This proof is similar to Lemma A.1. $\qquad\square$

By Lemma B.6, we derive the characteristic function of $\tilde{s}X$, as shown in Lemma B.7. In the following derivation, we will slightly abuse the notation by using $\tilde{\mu}$ and $\tilde{\sigma}^2$, i.e., $\tilde{\mu} = \tilde{\mu}_{1/2}$ and $\tilde{\sigma}^2 = \tilde{\sigma}_{1/2}^2$.

**Lemma B.7.** *The characteristic function of $\tilde{s}X$ under $l_{1/2}$ norm is as follows:*

$$\varphi_{\tilde{s}X}(x) = \frac{1}{2(1 - \Phi(\frac{-\tilde{\mu}}{\tilde{\sigma}}))} \exp\left(\frac{-\tilde{\mu}^2 + (\tilde{\mu} - \sqrt{-2ix}\tilde{\sigma}^2)^2}{2\tilde{\sigma}^2}\right) \text{erfc}\left(\frac{\sqrt{-2ix}\tilde{\sigma}^2 - \tilde{\mu}}{\sqrt{2}\tilde{\sigma}}\right) \quad (-\infty < x < +\infty)$$

*where $\tilde{\mu} = \frac{ms^{1/2}}{n}$ and $\tilde{\sigma}^2 = m\sigma_{1/2}^2$; $\text{erfc}(t) = \frac{2}{\sqrt{\pi}}\int_t^{+\infty} \exp(-x^2)dx$ $(-\infty < t < +\infty)$ is the complementary error function.*

*Proof.* According to Eqn. 21, we know the PDF of $\tilde{s}$ under $l_{1/2}$ norm. By applying Lemma B.7, we have:

$$\varphi_{\tilde{s}X}(x) = \frac{1}{\sqrt{2\pi}\tilde{\sigma}} \int_0^{+\infty} \frac{\exp(-\sqrt{-2ixy} - \frac{(\sqrt{y}-\tilde{\mu})^2}{2\tilde{\sigma}^2})}{2\sqrt{y}(\Phi(a_2; \tilde{\mu}, \tilde{\sigma}^2) - \Phi(a_1; \tilde{\mu}, \tilde{\sigma}^2))} dy$$

$$= \frac{1}{\sqrt{2\pi}\tilde{\sigma}} \int_0^{+\infty} \frac{\exp(-\sqrt{-2ix}y^2 - \frac{(y-\tilde{\mu})^2}{2\tilde{\sigma}^2})}{\Phi(a_2; \tilde{\mu}, \tilde{\sigma}^2) - \Phi(a_1; \tilde{\mu}, \tilde{\sigma}^2)} dy$$

$$= \frac{1}{\sqrt{2\pi}\tilde{\sigma}} \int_0^{+\infty} \frac{\exp(-\frac{(y^2 - 2\tilde{\mu}y + 2\sqrt{-2ix}\tilde{\sigma}^2 y + \tilde{\mu}^2)}{2\tilde{\sigma}^2})}{\Phi(a_2; \tilde{\mu}, \tilde{\sigma}^2) - \Phi(a_1; \tilde{\mu}, \tilde{\sigma}^2)} dy$$

$$= \frac{1}{\sqrt{2\pi}\tilde{\sigma}} \int_0^{+\infty} \frac{\exp(\frac{-\tilde{\mu} + (\tilde{\mu} - \sqrt{-2ix}\tilde{\sigma}^2)^2}{2\sigma^2}) \cdot \exp(-\frac{(y - (\tilde{\mu} - \sqrt{-2ix}\tilde{\sigma}^2))^2}{2\tilde{\sigma}^2})}{\Phi(a_2; \tilde{\mu}, \tilde{\sigma}^2) - \Phi(a_1; \tilde{\mu}, \tilde{\sigma}^2)} dy$$

$$= \frac{1}{2(1 - \Phi(\frac{-\tilde{\mu}}{\tilde{\sigma}}))} \cdot \exp\left(\frac{-\tilde{\mu} + (\tilde{\mu} - \sqrt{-2ix}\tilde{\sigma}^2)^2}{2\sigma^2}\right) \cdot \text{erfc}\left(\frac{\sqrt{-2ix}\tilde{\sigma}^2 - \tilde{\mu}}{\sqrt{2}\tilde{\sigma}}\right)$$

where $\tilde{\mu} = \frac{ms^{1/2}}{n}$, $\tilde{\sigma}^2 = m\sigma_{1/2}^2$, $a_2 = \infty$ and $a_1 = 0$. Hence we prove this Lemma $\qquad\square$

Given the characteristic function $\varphi_{\tilde{s}X}(x)$ under the $l_{1/2}$ norm, the probability density function (denoted by $l_{\tilde{s}X}(t)$) of $\tilde{s}X$ can be obtained by Eqn. 9. As a result, the collision probability for a pair $(\boldsymbol{v}, \boldsymbol{u})$ is computed as that of Theorem A.3, in which $f_{|\tilde{s}X|}(t)$ is replaced with $l_{|\tilde{s}X|}(t)$.

Similar to Theorem A.4, the following theorem gives the asymptotic behavior of the characteristic function of $\tilde{s}X$ under the $l_{1/2}$ norm, that is:

**Theorem B.8.**

$$\lim_{m\to+\infty} \frac{\varphi_{\tilde{s}X}(x)}{\exp(-\sqrt{\frac{-2im^2sx}{n^2}})} = 1$$

where $\exp(-\sqrt{-2im^2sx/n^2})$ is the characteristic function of Lévy distribution $\mathcal{L}(0, \frac{ms^{1/2}}{n})$.

*Proof.* Given that $\tilde{\mu} = \tilde{\mu}_{1/2} = \frac{ms^{1/2}}{n}$ and $\tilde{\sigma}^2 = m\sigma_{1/2}^2$. Let $\sigma^2 = \sigma_{1/2}^2$. According to Lemma B.7, the characteristic function $\varphi_{\tilde{s}X}(x)$ can be written as:

$$\varphi_{\tilde{s}X}(x) = \frac{1}{2(1-\Phi(\frac{-\sqrt{m}s^{1/2}}{n\sigma}))} \exp(-im\sigma^2 x - \sqrt{-2im^2sx/n^2})\mathrm{erfc}(\frac{\sqrt{-2ix}m\sigma^2 - ms^{1/2}/n}{\sqrt{2}m\sigma})$$

Let $a = \frac{s^{1/2}}{\sqrt{2}n\sigma} > 0$ and $b = \sqrt{-i}\sigma > 0 \Rightarrow b^2 = -i\sigma^2$, where we focus on the real part of $b$. Using the known fact that $\Phi(x) = \frac{1}{2}\mathrm{erfc}(-\frac{x}{\sqrt{2}})$, we can simplify $\varphi_{\tilde{s}X}(x)$ as follows:

$$\varphi_{\tilde{s}X}(x) = \exp(b^2 mx - \sqrt{-2im^2sx/n^2}) \cdot \frac{\mathrm{erfc}(b\sqrt{mx} - a\sqrt{m})}{2 - \mathrm{erfc}(a\sqrt{m})}$$

Next, we define $g(x) = \frac{\varphi_{\tilde{s}X}(x)}{\varphi_{sX}(x)}$, where $\varphi_{sX}(x) = \exp(-\sqrt{-2im^2sx/n^2})$, which is the characteristic function of the Lévy distribution $\mathcal{L}(0, \frac{ms^{1/2}}{n})$. Thus, $g(x)$ can be written as:

$$g(x) = \exp(b^2 mx) \cdot \frac{\mathrm{erfc}(b\sqrt{mx} - a\sqrt{m})}{2 - \mathrm{erfc}(a\sqrt{m})}$$

To prove $\varphi_{\tilde{s}X}(x) = \varphi_{sX}(x)$, we need to show that as $m \to +\infty$, $g(x) = 1$. Obviously, $x \le O(m^{-2})$, we have $\sqrt{mx} \le O(m^{-1/2})$ and $mx \le O(m^{-1})$. Thus, for any constant $b \in \mathbb{R}^+$, the following holds:

$$\lim_{m\to+\infty} \exp(b^2 mx) = 1$$

On the other hand, we have:

$$\mathrm{erfc}(b\sqrt{mx} - a\sqrt{m}) \sim \mathrm{erfc}(-a\sqrt{m}) \quad m \to +\infty$$

Since $b\sqrt{mx} - a\sqrt{m} \sim -a\sqrt{m}$ as $m \to +\infty$ and $\mathrm{erfc}(-x) \sim 2 - \frac{\exp(-x^2)}{\sqrt{\pi}x}$ as $x \to +\infty$, we obtain:

$$\lim_{m\to+\infty} \frac{\mathrm{erfc}(b\sqrt{mx} - a\sqrt{m})}{2 - \mathrm{erfc}(a\sqrt{m})} = \frac{\lim_{m\to+\infty}\mathrm{erfc}(b\sqrt{mx} - a\sqrt{m})}{\lim_{m\to+\infty}\mathrm{erfc}(-a\sqrt{m})} = 1$$

Then, we have:

$$g(x) = \lim_{m\to+\infty} \exp(b^2 mx) \cdot \lim_{m\to+\infty} \frac{\mathrm{erfc}(b\sqrt{mx} - a\sqrt{m})}{\mathrm{erfc}(-a\sqrt{m})} = 1$$

Hence, we prove this Theorem. $\qquad \square$

Theorem B.8 means that $l_{\tilde{s}X}(t)$ is asymptotically equivalent to $\mathcal{L}(0, \frac{ms^{1/2}}{n})$. Then the following Corollary is naturally yielded:

**Corollary B.9.** $l_{\tilde{s}X}(t) \sim$ *the PDF of* $\mathcal{L}(0, \frac{ms^{1/2}}{n})$ *as m tends to infinity.*

## C ADDITIONAL EXPERIMENTS

In this section we present more description about datasets, parameter setting and additional experiments for three machine learning tasks. All experiments for nearest neighbor search and outlier detection are carried out on a server with six-cores Intel(R), i7-8750H @ 2.20GHz CPU and 32 GB RAM, in Ubuntu 20.04. Experiments for neural network training are carried out on a server with fourteen-cores Intel(R), i7-12700H @ 2.30GHz CPU and 64 GB RAM, in Ubuntu 20.04.

### C.1 OUTLIER DETECTION

**Baseline:** Anomaly detection is a critical task in data analysis, which aims to identify instances or patterns that deviate from expected behavior. For anomaly detection task, there are two kinds of methods, i.e., supervised and unsupervised methods. Unsupervised methods are more preferred in practice due to their ability to adapt to changing data distributions without requiring label information. Existing unsupervised anomaly detection approaches often require storing the entire dataset, leading to poor computational and memory requirements, particularly as data volume increases. To overcome these limitations, researchers propose Arrays of (locality-sensitive) Count Estimators (ACE) in Luo & Shrivastava (2018), a novel anomaly detection algorithm, for high-speed streaming data and constrained memory environments.

ACE is an efficient anomaly detection data structure, which is composed of multiple locality-sensitive hash tables. These hash tables are used to estimate counts of collision and detect anomalies by performing hash lookup. Specifically, a hash code $H(\boldsymbol{v}) = (h_1(\boldsymbol{v}), h_2(\boldsymbol{v}), \ldots, h_k(\boldsymbol{v})$ of $k$ bits is computed using $k$ independent LSH functions. Then $L$ groups of hash functions $H_i(\cdot), i = \{1, \cdots, L\}$ are drawn independently and uniformly at random from the LSH family. Instead of constructing $L$ hash tables, ACE constructs $L$ short arrays, $A_i$, of size $2^k$ each initialized with zeros. Given any observed element $\boldsymbol{v} \in \mathbf{D}$, ACE increments the count of the corresponding counter $H_i(\boldsymbol{v})$ in array $A_i$ for all $i$.

To decide if $\boldsymbol{u}$ is an outlier, ACE computes the average of all the counters for $\forall i \in \{1, 2, \ldots L\}$, i.e., $\hat{S}(\boldsymbol{u}, \mathbf{D}) = \frac{1}{L}\sum_{i=1}^{L} A_i[H_i(\boldsymbol{u})]$. $\boldsymbol{u}$ is reported as anomaly if the estimated score $\hat{S}(\boldsymbol{u}, \mathbf{D})$ is less than $\mu - \sigma$, where $\mu = \frac{1}{N}\sum_{i=1}^{N}\hat{S}(\boldsymbol{v}_i, \mathbf{D})$ is the mean of the scores over all $\boldsymbol{v} \in \mathbf{D}$ and $\sigma$ is the standard deviation. To evelute FastLSH and ACHash in the outlier detection task, we only need to replace the hash functions used in ACE with FastLSH and ACHash. The corresponding methods are termed as FastACE (FastLSH + ACE) and ACHashACE (ACHash +ACE), respectively.

**Evaluation Metrics:** Similar to Luo & Shrivastava (2018), the following five performance measures are used: 1) outliers reported, i.e., the number of outliers detected by the algorithm. 2) correctly reported, i.e., the number of truth outliers in the outliers reported. 3) outlier missed, i.e., the remaining number of truth outliers in all truth outliers after outliers reported having been detected. 4) the execution time taken to report the outliers. 5) the speedup over ACE.

**Datasets:** We choose three widely used real-world benchmark datasets for anomaly detection, the statistics of which are presented in Table 4.

Table 4: Statistics of the datasets

| Datasets | # of Instances | # of Outliers | Dimension |
|---|---|---|---|
| Statlog Shuttle | 34,987 | 879 | 9 |
| a9a | 48,842 | 7841 | 123 |
| Musk | 6,598 | 97 | 166 |

Statlog Shuttle[4]: It is the dataset of radiator positions in a NASA space shuttle with 9 attributes designed for supervised anomaly detection. The dataset includes 34987 instances with 879 anomalies.

a9a[5]: It is obtained from UCI Adult dataset, which predicts whether income exceeds 50K dollars per year. The dataset contains 48842 instances with each having 123 features. There are two classes

---

[4]https://archive.ics.uci.edu/ml/datasets/Statlog+(Shuttle)
[5]https://www.csie.ntu.edu.tw/ cjlin/libsvmtools/datasets/binary.html

Table 5: Statistics of Datasets

| Datasets | Feature Dim | Feature Sparsity | Label Dim | Training Size | Testing Size |
|---|---|---|---|---|---|
| Delicious-200K | 782,585 | 0.038% | 205,443 | 196,606 | 100,095 |
| Amazon-670K | 135,909 | 0.055% | 670,091 | 490,449 | 153,025 |

of labels denoted as -1 and 1, where 1 is the income exceeding 50K, while the other is lower than 50K.

Musk[6]: It is the set of 102 molecules of which 39 molecules are judged by human experts to be musks and the remaining 93 molecules are non-musks. These molecules are to generate 6598 conformations with each conformation contains 166 features.

**Parameter Settings:** We use $k = 15$ and $L = 50$ for ACE, ACHashACE and FastACE as in Luo & Shrivastava (2018). For FastACE, $m$ is set to 3 for Statlog Shuttle, and $m = 30$ for the other two datasets. For ACHashACE, the sampling ratio is set to the default 0.25 Dasgupta et al. (2011).

### C.2 Neural Network Training

**Baseline:** Deep Learning (DL) has drawn a lot of attention in recent years, transforming fields such as computer vision, natural language processing and speech recognition. Training a DL mode from scratch demands massive computing resources. While dedicated hardware offers accelerated performance in matrix multiplication, it comes with risks and limitations, including substantial investment requirements and the possibility of becoming obsolete with advancements in algorithms. To this end, SLIDE (Sub-LInear Deep learning Engine) Chen et al. (2020) is proposed to train DL models using only commodity CPUs by exploiting adaptive sparsity in neural networks, especially in large fully connected neural networks.

SLIDE is a novel deep learning engine that integrates randomized algorithms (LSH) and multi-core parallelism. It achieves training speeds faster than using Tensorflow on GPU, exclusively utilizing a CPU on large-scale recommendation datasets Bhatia et al. (2016); Mittal et al. (2022; 2021). Briefly SLIDE works as follows. In a fully connected neural network, each layer consists of a list of neurons and a set of LSH sampling hash tables. During network initialization, the weights are randomly initialized, and $k \times L$ LSH hash functions are set up along with $L$ hash tables for each layer. These hash functions compute hash codes $h_l(w_l^a)$ for the weight vectors of neurons in each layer, where $h_l$ represents the hash function in layer $l$ and $w_l^a$ is the weights for the $a^{th}$ neuron in layer $l$. The neuron's id $a$ is stored in the hash buckets determined by the LSH function $h_l(w_l^a)$. In SLIDE, rather than calculating all activations in each layer, the input to each layer $v_l$ is passed through hash functions to compute $h_l(v_l)$. These hash codes act as queries to retrieve the ids of active (or sampled) neurons from corresponding buckets in the hash tables. For each training data instance, the neuron backpropagates partial gradients (using error propagation) exclusively to active neurons in previous layers through the connected weights.

**Evaluation Metrics:** We report the classification accuracy, the end-to-end training time and the number of iterations as in Chen et al. (2020).

**Datasets:** We employ two large real datasets, Delicious-200K and Amazon-670K, from the Extreme Classification Repository Bhatia et al. (2016), and the statistics of the two datasets are presented in Table 5. Delicious-200K dataset is a subsampled dataset generated from a vast corpus of almost 150 million bookmarks from Social Bookmarking Systems. Amazon-670K dataset is a product to product recommendation dataset with 670K labels.

**Parameter Settings:** For both datasets, a standard fully connected neural network with one hidden layer of size 128 and a batch size of 128 is employed. All algorithms are executed until convergence. To evaluate the performance of SLIDE, FastSLIDE (FastLSH + SLIDE) and ACHashSLIDE (ACHash + SLIDE), we utilize the same optimizer, Adam, while adjusting the initial step size from $1e^{-5}$ to $1e^{-3}$ to ensure better convergence across all experiments. In SLIDE, we particularly focus on maintaining hash tables for the last layer, which is often a computational bottleneck in the

---
[6]https://archive.ics.uci.edu/dataset/75/musk+version+2

Table 6: Statistics of Datasets

| Datasets | # of Points | # of Queries | Dimension |
|---|---|---|---|
| Sun | 69106 | 200 | 512 |
| Cifar | 50000 | 200 | 512 |
| Audio | 53387 | 200 | 192 |
| Trevi | 99000 | 200 | 4096 |
| Notre | 333000 | 200 | 128 |
| Sift | 1000000 | 200 | 128 |
| Gist | 1000000 | 200 | 960 |
| Deep | 1000000 | 200 | 256 |
| Ukbench | 1000000 | 200 | 128 |
| Glove | 1192514 | 200 | 100 |
| ImageNet | 2340000 | 200 | 150 |
| Random | 1000000 | 200 | 100 |

models. In terms of LSH settings, we set $k = 9$ and $L = 50$ for Decilious-200K and $k = 8$ and $L = 50$ for Amazon-670K. The hash tables are updated initially every $N_0 = 50$ iterations and then exponentially decayed. The sampling ratio of FastSLIDE is set to 0.15 and 0.07 for Delicious and Amazon, respectively. For ACHashSLIDE, the sampling ratio is set to the default 0.25 Dasgupta et al. (2011).

### C.3 FASTLSH VS. E2LSH FOR NEAREST NEIGHBOR SEARCH

**Baseline:** Nearest neighbor search (NNS) is an essential problem in machine learning, which has numerous applications such as face recognition, information retrieval and duplicate detection. The purpose of nearest neighbor search is to find the point in the dataset $\mathbf{D} = \{\boldsymbol{v}_1, \boldsymbol{v}_2, \ldots, \boldsymbol{v}_N\}$ that is most similar (has minimal distance) to the given query $\boldsymbol{u}$. For low dimensional spaces ($<10$), popular tree-based index structures such as KD-tree Bentley (1975), SR-tree Katayama & Satoh (1997), etc. deliver exact answers and provide satisfactory performance. For high dimensional spaces, however, these index structures suffer from the well-known curse of dimensionality, that is, their performance is even worse than that of linear scans Samet (2006). To address this issue, one feasible way is to use approximate nearest neighbor (ANN) search by trading accuracy for efficiency Kushilevitz et al. (1998).

The canonical LSH index structure (E2LSH) for ANN search is built as follows. A hash code $H(\boldsymbol{v}) = (h_1(\boldsymbol{v}), h_2(\boldsymbol{v}), \ldots, h_k(\boldsymbol{v})$ is computed using $k$ independent LSH functions (i.e., $H(\boldsymbol{v})$ is the concatenation of $k$ elementary LSH codes). Then a hash table is constructed by adding the 2-tuple $\langle H(\boldsymbol{v}), id\ of\ \boldsymbol{v} \rangle$ into corresponding bucket. To boost accuracy, $L$ groups of hash functions $H_i(\cdot), i = 1, \cdots, L$ are drawn independently and uniformly at random from the LSH family, resulting in $L$ hash tables.

To answer a query $\boldsymbol{u}$, one need to first compute $H_1(\boldsymbol{u}), \cdots, H_L(\boldsymbol{u})$ and then search all these $L$ buckets to obtain the combined set of candidates. Then, all points in the candidate set are evaluated against the query and the most similar points are returned. There exists two ways (approximate and exact) to process these candidates. In the approximate version, no more than $3L$ points in the candidate set are evaluated. The LSH theory ensures that the $(c, R)$-NN is found with a constant probability. In practice, however, the exact one is widely used since it offers better accuracy at the cost of evaluating all points in the candidate set Datar et al. (2004). The search time consists of both the hashing time and the time taken to prune the candidate set Datar et al. (2004); Andoni (2005). In many cases, nearest neighbor search is just one component of a larger application that involves other approximations. As a result, using approximate neighbors instead of exact ones often leads to minimal performance loss. Therefore, we use the exact method to process a query similar to Datar et al. (2004); Andoni (2005).

**Evaluation Metrics:** To evaluate the performance of FastLSH and baselines, we present the following metrics: 1) recall, i.e., the fraction of near neighbors that are actually returned; 2) the average running time to report the near neighbors for each query; 3) the time taken to compute hash functions; 4) the end-to-end index construction time.

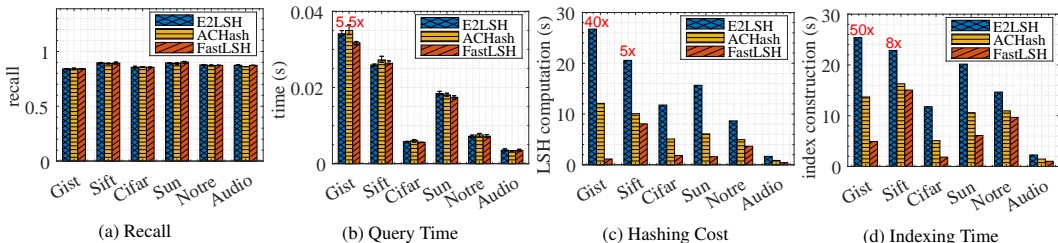

Figure 4: Comparison of recall, average query time, LSH computation and index construction time.

**Datasets:** 11 publicly available high-dimensional real datasets and one synthetic dataset are experimented with Li et al. (2019), the statistics of which are listed in Table 6. Sun[7] is the set of containing about 70k GIST features of images. Cifar[8] is denoted as the set of 50k vectors extracted from Tiny-Image. Audio[9] is the set of about 50k vectors extracted from DARPA TIMIT. Trevi[10] is the set of containing around 100k features of bitmap images. Notre[11] is the set of features that are Flickr images. Sift[12] is the set of containing 1 million SIFT vectors. Gist[13] is the set that is consist of 1 million image vectors. Deep[14] is the set of 1 million vectors that are deep neural codes of natural images obtained by convolutional neural network. Ukbench[15] is the set of vectors containing 1 million features of images. Glove[16] is the set of about 1.2 million feature vectors extracted from Tweets. ImageNet[17] is the set of data points containing about 2.4 million dense SIFT features. Random is the synthetic dataset containing 1 million randomly selected vectors in a unit hypersphere.

**Parameter Settings:** For the same dataset and target recall, we use identical $k$ (number of hash functions per table) and $L$ (number of hash tables) for fairness. Thus, three algorithms take the same space occupations. $m$ is set to 30 throughout all experiments for FastLSH. To achieve optimal performance, the sampling ratio for ACHash is set to the default 0.25 Dasgupta et al. (2011). Table 7 reports the parameters ($k$, $L$ and bucket width $w$) for different target recall illustrated in Figure 5, where $w$ is the bucket width of E2LSH, $\tilde{w}$ and $w'$ are those of FastLSH and ACHash respectively.

**Results and Discussion:** In this set of experiments, we are intended to show that FastLSH can reduce the index construction time significantly and achieve almost the same recall and query time as other LSH-based algorithms in the meantime.

We first compare the performance among E2LSH, ACHash and FastLSH for target recall around 0.9. The recall, average query time and LSH computation time for Gist, Sift, Cifar, Sun, Notre and Audio are illustrated in Figure 4 (a), (b) and (c). It is easy to see that FastLSH and E2LSH achieve comparable query performance and answer accuracy as plotted in Figure 4 (a) and (b). Due to lack of theoretical guarantee, ACHash performs slightly worse than FastLSH and E2LSH in most cases w.r.t query efficiency. As shown in Figure 4 (c), the LSH computation time of FastLSH is significantly superior to E2LSH and ACHash. For example, FastLSH obtains around 24x speedup over E2LSH and runs 11x times faster than ACHash on *Gist*. For ACHash, the fixed sampling ratio and overhead in Hadamard transform make it inferior to FastLSH. Note that because the query time, hashing cost and index construction time for different datasets varies greatly among datasets, we use 40x and etc. in the plots to indicate that the actual time is 40 times as much as the one shown in the plots.

---

[7]http://groups.csail.mit.edu/vision/SUN/

[8]http://www.cs.toronto.edu/ kriz/cifar.html

[9]http://www.cs.princeton.edu/cass/audio.tar.gz

[10]http://phototour.cs.washington.edu/patches/default.htm

[11]http://phototour.cs.washington.edu/datasets/

[12]http://corpus-texmex.irisa.fr

[13]https://github.com/aaalgo/kgraph

[14]https://yadi.sk/d/I_yaFVqchJmoc

[15]http://vis.uky.edu/ stewe/ukbench/

[16]http://nlp.stanford.edu/projects/glove/

[17]http://cloudcv.org/objdetect/

Table 7: Parameters of E2LSH, FastLSH and ACHash

| Datasets | recall | $k$ | $L$ | $w$ | $\widetilde{w}$ | $w'$ |
|---|---|---|---|---|---|---|
| Cifar | lowest | 10 | 40 | 0.0175 | 0.0041 | 0.138 |
| | median | 10 | 40 | 0.0185 | 0.0042 | 0.15 |
| | highest | 10 | 40 | 0.0195 | 0.00428 | 0.156 |
| Sun | lowest | 8 | 45 | 4200 | 980 | 34000 |
| | median | 8 | 45 | 4550 | 1050 | 36000 |
| | highest | 8 | 45 | 5050 | 1120 | 39000 |
| Gist | lowest | 10 | 105 | 2.5 | 0.435 | 0.45 |
| | median | 10 | 105 | 2.75 | 0.45 | 0.48 |
| | highest | 10 | 105 | 2.9 | 0.5 | 0.52 |
| Trevi | lowest | 8 | 105 | 11.8 | 1 | 370 |
| | median | 8 | 105 | 12.8 | 1.1 | 400 |
| | highest | 8 | 105 | 14 | 1.2 | 435 |
| Audio | lowest | 6 | 25 | 7000 | 2680 | 47500 |
| | median | 6 | 25 | 7700 | 2900 | 50000 |
| | highest | 6 | 25 | 8700 | 3398 | 61000 |
| Notre | lowest | 6 | 35 | 29 | 13.8 | 116 |
| | median | 6 | 35 | 31.5 | 15 | 125 |
| | highest | 6 | 35 | 35 | 16.6 | 134 |
| Glove | lowest | 8 | 79 | 0.84 | 0.455 | 3 |
| | median | 8 | 105 | 0.84 | 0.455 | 3 |
| | highest | 9 | 150 | 0.91 | 0.495 | 3 |
| Sift | lowest | 8 | 45 | 36 | 17 | 145 |
| | median | 8 | 45 | 39.5 | 19 | 160 |
| | highest | 8 | 45 | 42 | 20 | 165 |
| Deep | lowest | 8 | 60 | 0.0565 | 0.0189 | 0.45 |
| | median | 8 | 80 | 0.0565 | 0.0189 | 0.45 |
| | highest | 8 | 105 | 0.0565 | 0.0189 | 0.45 |
| Random | lowest | 10 | 108 | 0.5 | 0.275 | 1.85 |
| | median | 10 | 108 | 0.54 | 0.285 | 2 |
| | highest | 10 | 108 | 0.6 | 0.295 | 2.2 |
| Ukbench | lowest | 8 | 55 | 20.5 | 9.5 | 72 |
| | median | 8 | 75 | 20.5 | 9.5 | 72 |
| | highest | 8 | 105 | 20.5 | 9.5 | 80 |
| ImageNet | lowest | 8 | 105 | 0.36 | 0.162 | 0.159 |
| | median | 8 | 105 | 0.4 | 0.175 | 0.17 |
| | highest | 8 | 105 | 0.465 | 0.203 | 0.2 |

We also plot the recall v.s. average query time curves by varying target recalls to obtain a complete picture of FastLSH in Figure 5. The empirical results demonstrate that FastLSH performs almost the same in terms of answer accuracy, query efficiency and space occupation as E2LSH. Again, ACHash is slightly inferior to the others in most cases.

The end-to-end speedup in the index construction time is shown in Figure 4 (d). Thanks to the significant reduction in hashing cost, the time spent in building the index decreases by up to a factor of 20. Besides hashing, the procedure of index construction consists of some other operations such as hash table initialization and linked list maintainance, which cannot be accelarated. Thus, the end-to-end latency in index construction decreases not as much as the hashing cost.

In sum, FastLSH is on par with E2LSH (with provable LSH property) in terms of answer accuracy and query efficiency and marginally superior to ACHash (without provable LSH property), while significantly reducing the cost of hashing.

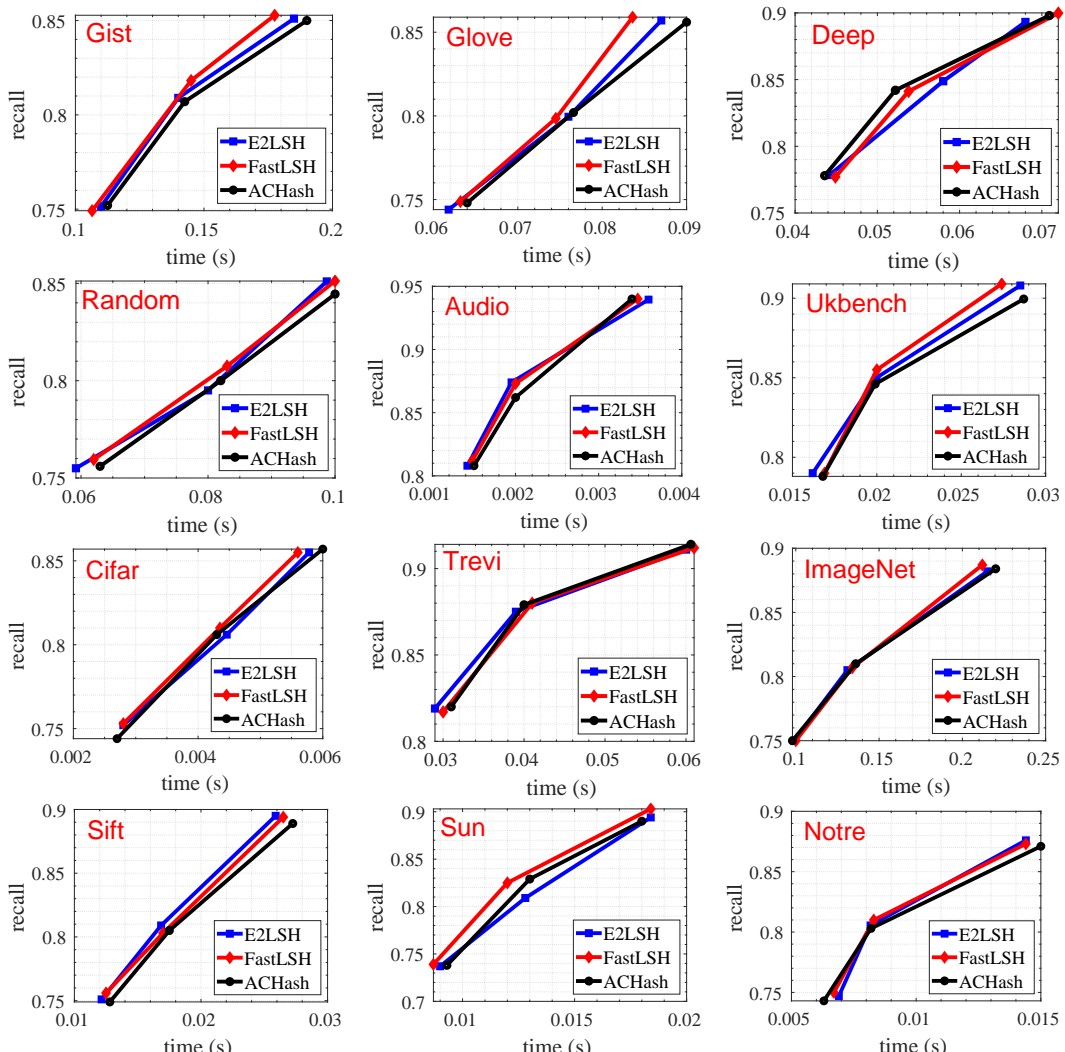

Figure 5: Recall vs. Average query time

## C.4 FASTLSH VS. MULTI-PROBE LSH IN NEAREST NEIGHBOR SEARCH

**Baseline:** MPLSH[18] Lv et al. (2007) is a variant of vanilla LSH (E2LSH) designed for ANN search, which uses a heuristic probe sequence to search multiple buckets that contain the NNs of given query with high probability. Compared with E2LSH, MPLSH provides better space and time efficiency, i.e., it achieves the reduction in memory usage by using less hash functions ($k \times L$) and the same recall with lower query response time.

**Metrics and Datasets:** The same as those in evaluating FastLSH and E2LSH.

**Parameter Settings:** For fairness, we set the same parameters $k$, $L$ and probe sequence for FastLSH and MPLSH to obtain the target recall. The parameters are presented in Table 8, where $w$ is the bucket width of MPLSH, $\tilde{w}$ and $w'$ are that of FastLSH and ACHash respectively; $m$ is the number of sampled dimensions for FastLSH and the sampling ratio for ACHash is set to the defaulst 0.25 as in Dasgupta et al. (2011).

**Results and Discussion:** In this set of experiments, we achieve around 0.9 target recall for MPLSH, ACHash and FastLSH over all tested datasets. The actual recall, query time, hashing time and speedup in hashing are shown in Figure 6. We can observe the same trend for MPLSH as with

---

[18]https://lshkit.sourceforge.net/index.html

Table 8: Parameters of MPLSH, FastLSH and ACHash

| Datasets | recall | $k$ | $L$ | $w$ | $\tilde{w}$ | $w'$ | $m$ |
|---|---|---|---|---|---|---|---|
| Cifar | 0.85 | 10 | 5 | 0.65 | 0.15 | 0.16 | 30 |
| Sun | 0.8 | 6 | 15 | 99370 | 19500 | 18000 | 30 |
| Gist | 0.9 | 15 | 25 | 3.9 | 1.07 | 0.355 | 90 |
| Trevi | 0.9 | 10 | 25 | 3500 | 1100 | 225 | 512 |
| Audio | 0.88 | 10 | 5 | 166000 | 80000 | 37000 | 60 |
| Notre | 0.88 | 6 | 10 | 420 | 110 | 93 | 30 |
| Glove | 0.88 | 15 | 25 | 16 | 8.1 | 5.2 | 50 |
| Sift | 0.9 | 15 | 25 | 1000 | 256 | 230 | 30 |
| Deep | 0.9 | 15 | 25 | 2.5 | 0.57 | 0.375 | 30 |
| Random | 0.9 | 15 | 25 | 8 | 4.5 | 4.3 | 50 |
| Ukbench | 0.87 | 6 | 10 | 220 | 60 | 52 | 30 |
| ImageNet | 0.9 | 10 | 25 | 0.65 | 0.28 | 0.18 | 75 |

E2LSH, as shown in Figure 4. Particularly, FastLSH achieves around the same recall for each dataset with the same or even much better query time than MPLSH and ACHash. For half of the datasets, $m = 30$ offers nice performance for FastLSH, whereas the others need much larger $m$ to obtain the target recall. This might be attributed to the data-dependent nature of FastLSH and the probing heuristics used by MPLSH. The performance of ACHash is inferior to FastLSH due to lack of provable LSH property. The experimental results indicate that FastLSH performs well for all datasets and obtains up to 12x and 3x speedup in hashing over MPLSH and ACHash, respectively. The end-to-end index construction time is presented in Figure 6 (d). Likewise, efficient hashing translates to reduced indexing construction time, leading to up to 10x speedup in building the index.

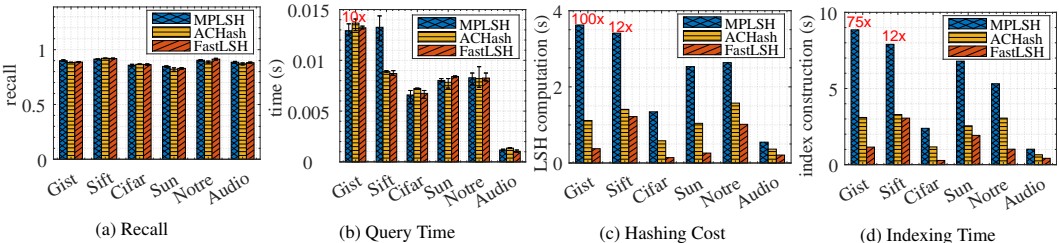

Figure 6: Comparison of recall, average query time, LSH computation and index construction time.

## C.5 FastLSH Handles Sparse Data

Actually, FastLSH does work on sparse vector. The reasons and empirical evidence are as follows.

Recall that FastLSH consists of random sampling and random projection. FastLSH first performs $m$ random sampling operations. The probability of selecting at least one non-zero (significant) element during the process of random sampling is $p_1 = 1 - (1 - p)^m$, where $p$ is the proportion of non-zero elements in the $n$-dimensional vector. And then FastLSH performs random projection. Since the hash code of FastLSH is a fingerprint concatenated from $k$ hashes, the probability of selecting at least one non-zero (significant) element under $k$ hashes is $p_r = 1 - (1 - p_1)^k$. As a quick example, if $p = 0.01$, $m = 30$ and $k = 8$, then $p_r = 0.95$, meaning that the hash code contains the information of the significant elements with a high probability of 0.95. Therefore, FastLSH can effectively handle sparse data and is also well-suited for dense data, particularly in cases where many coordinate values are nearly identical.

We chose the MNIST[19] dataset to verify this claim. The dimension of MNIST is 784, with only around 2.6% non-zero elements. When we set $m = 30$ for FastLSH, the experimental results of FastLSH, E2LSH and ACHash under different $k$ and $L$ are shown in Table 9. From the table, it can

---

[19]http://yann.lecun.com/exdb/mnist/

Table 9: Results of E2LSH, FastLSH and ACHash over MNIST under different parameters

| Methods | Recall | Time (s) | Hashing (s) | Indexing (s) | $k$ | $L$ | $w$ |
|---------|--------|----------|-------------|--------------|-----|-----|-----|
| E2LSH | 0.844 | 0.00541 | 9.818 | 11.302 | 8 | 50 | 0.18 |
| ACHash | 0.853 | 0.00547 | 5.151 | 6.683 | 8 | 50 | 92.5 |
| FastLSH | 0.8545 | 0.00482 | **0.671** | **2.301** | 8 | 50 | 0.0325 |
| E2LSH | 0.799 | 0.00460 | 7.703 | 8.875 | 8 | 40 | 0.18 |
| ACHash | 0.809 | 0.00465 | 3.911 | 5.063 | 8 | 40 | 92.5 |
| FastLSH | 0.816 | 0.00439 | **0.542** | **1.828** | 8 | 40 | 0.0325 |
| E2LSH | 0.741 | 0.00352 | 6.365 | 7.307 | 8 | 30 | 0.18 |
| ACHash | 0.750 | 0.00360 | 3.328 | 4.339 | 8 | 30 | 92.5 |
| FastLSH | 0.748 | 0.00315 | **0.417** | **1.369** | 8 | 30 | 0.0325 |

be seen that FastLSH and E2LSH obtain comparable query accuracy, meaning it works very well for sparse datasets. While for ACHash, the fixed sampling ratio and overhead in Hadamard transform make it inferior to FastLSH. At the mean time, FastLSH can achieve up to 14.6 and 7.7 times faster LSH computation compared with E2LSH and ACHash, respectively. As a result, FastLSH can significantly reduce the end-to-end index construction time.

Note that for E2LSH package that we used Andoni (2005), ultrahigh-dimensional vectors are assumed dense by default and all elements, regardless of whether it is zero or not, have to be computed. That is why such high speedup in hashing was achieved for FastLSH. In addition, for sparse ultrahigh-dimensional vectors, if the number of non-zero elements is still high after data preprocessing, FastLSH can be used to handle these non-zero elements; otherwise, we use the following Fast Johnson-Lindenstrauss (JL) transform.

Actually, for most practical applications, FastLSH is sufficient to handle sparse data as we have shown with the MINIST dataset. However, in the case of datasets being pathologically sparse, we can first apply the Fast JL transform to make the data dense similar to ACHash Dasgupta et al. (2011). Here the Fast JL transform refers to the Hadamard transform, which is a unitary transformation, meaning that it preserves the nearest neighbor relationships of data points after transformation. Then FastLSH performs random sampling and random projection for the dense data. Although these operations are similar to ACHash, FastLSH retains the provable LSH property, which ACHash does not.

### C.6   More Results for Comparison of Probability Density Curves

Lemma 4.8 and Fact 4.9 provide a principled approach to quantitatively analyze how $m$ affects the difference between FastLSH and the classic LSH in terms of $\epsilon$ and $\lambda$, given the dataset characteristics (the variance in the squared distances of coordinates for a pair of data items). By using this analytical tool, it is easy for practitioners to determine the trade-off between hashing time (how much $m$ is) and desired performance level (how close FastLSH is to the standard LSH).

To visualize the similarity, we plot $f_{\tilde{s}X}(t)$ for different $m$ under the maximum and minimum $\sigma$, and the PDF of $\mathcal{N}(0, \frac{ms^2}{n})$ in Figure 7 for all 12 datasets. The observations can be made from these figures: (1) the distribution of $\tilde{s}X$ matches very well with $\mathcal{N}(0, \frac{ms^2}{n})$ for small $\sigma$; (2) for large $\sigma$, $f_{\tilde{s}X}(t)$ differs only slightly from the PDF of $\mathcal{N}(0, \frac{ms^2}{n})$ for all $m$, indicating that $s$ is the dominating factor in $p(s, \sigma)$; (3) greater $m$ results in higher similarity between $f_{\tilde{s}X}(t)$ and $\mathcal{N}(0, \frac{ms^2}{n})$, implying that FastLSH can always achieve almost the same performance as E2LSH by choosing $m$ appropriately.

### C.7   Comparison of $\rho$ Curves

To further validate the LSH property of FastLSH, we compare the important parameter $\rho$ for FastLSH and E2LSH in the case of $m = 30$. $\rho$ is defined as the function of the approximation ratio $c$, i.e., $\rho(c) = log(1/p(s_1))/log(1/p(s_2))$, where $s_1 = 1$ and $s_2 = c$. Note that $\rho$ affects both the space and time efficiency of LSH algorithms. For $c$ in the range $[1, 20]$ (with increments

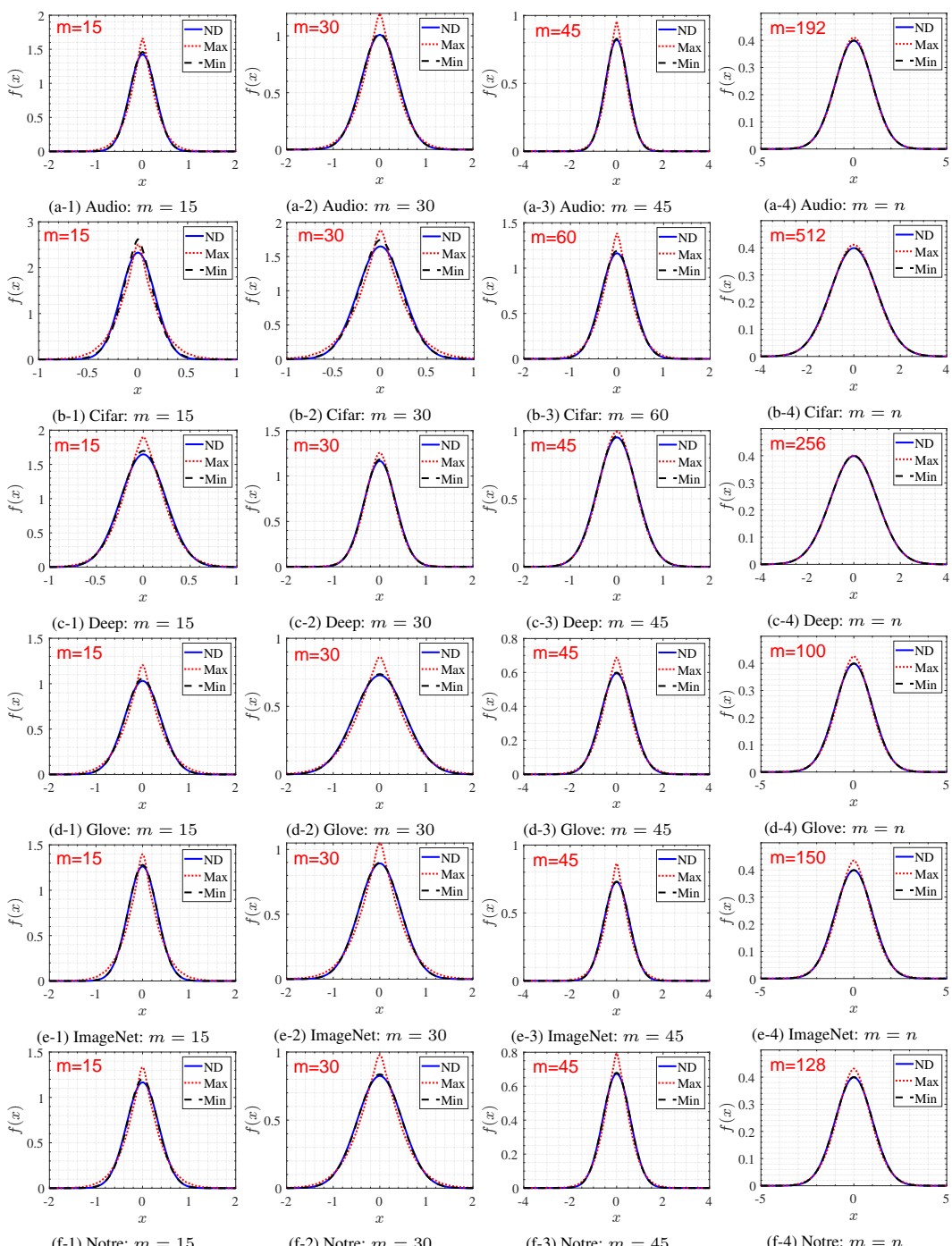

(a-1) Audio: $m = 15$  (a-2) Audio: $m = 30$  (a-3) Audio: $m = 45$  (a-4) Audio: $m = n$

(b-1) Cifar: $m = 15$  (b-2) Cifar: $m = 30$  (b-3) Cifar: $m = 60$  (b-4) Cifar: $m = n$

(c-1) Deep: $m = 15$  (c-2) Deep: $m = 30$  (c-3) Deep: $m = 45$  (c-4) Deep: $m = n$

(d-1) Glove: $m = 15$  (d-2) Glove: $m = 30$  (d-3) Glove: $m = 45$  (d-4) Glove: $m = n$

(e-1) ImageNet: $m = 15$  (e-2) ImageNet: $m = 30$  (e-3) ImageNet: $m = 45$  (e-4) ImageNet: $m = n$

(f-1) Notre: $m = 15$  (f-2) Notre: $m = 30$  (f-3) Notre: $m = 45$  (f-4) Notre: $m = n$

of 0.1), we calculate $\rho$ using *Matlab*, where the minimal and maximal $\sigma$ are collected for different $c$ ($s$). Plots of $\rho(c)$ under different bucket widths for 12 datesets are given in Figure 8 and Figure 9. Clearly, the $\rho(c)$ curve of FastLSH matches very well with that of E2LSH, verifying that FastLSH maintains almost the same LSH property with E2LSH even when $m$ is relatively small.

## C.8 EFFECTS OF $\epsilon$ AND $\lambda$

We list the values of $\epsilon$ and $\lambda$ for different $m$ over 12 datasets in Table 10, where $\epsilon$ and $\lambda$ are calculated using the maximum, mean and minimum $\sigma$, respectively. Recall that smaller $\epsilon$ and $\lambda$ are, FastLSH

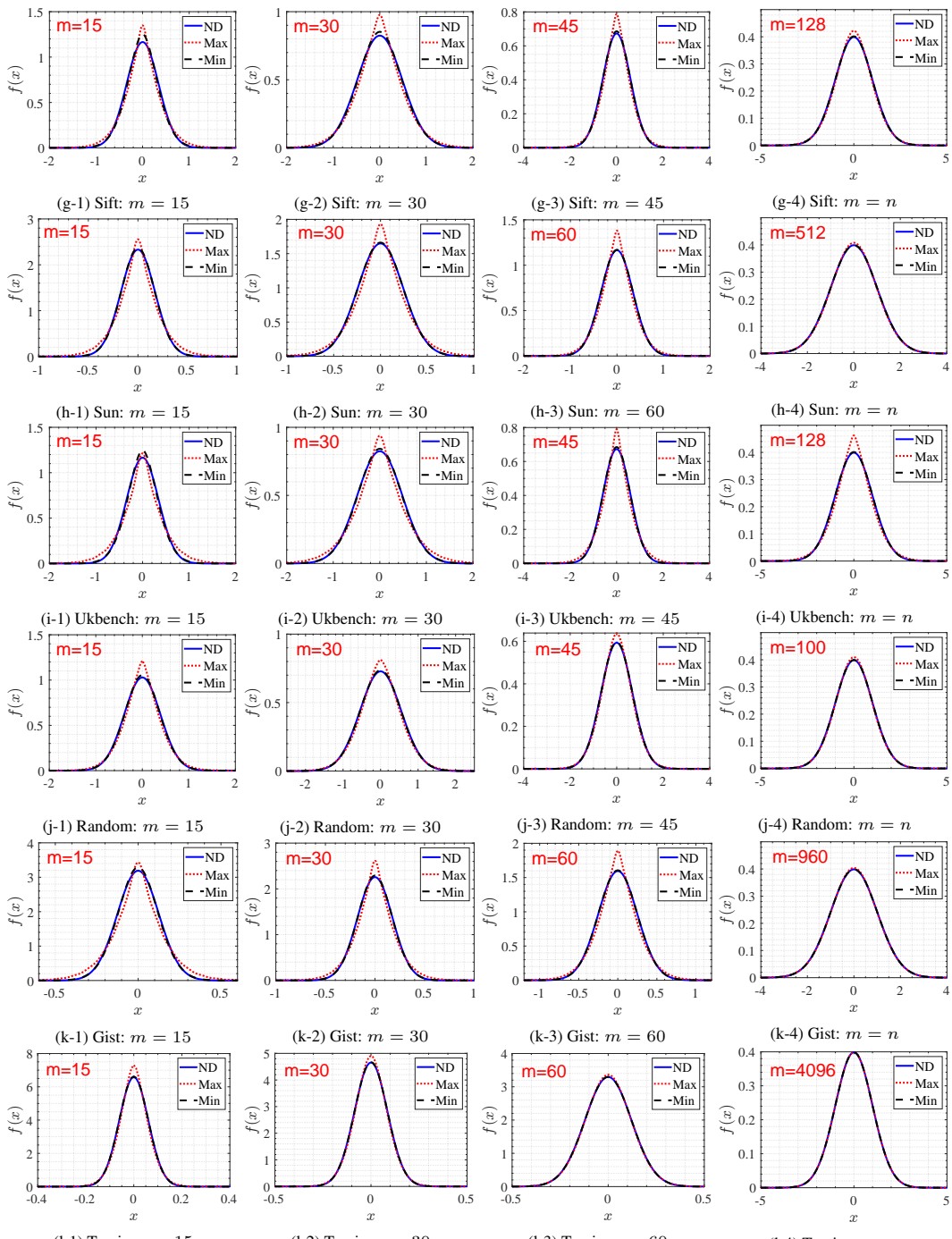

Figure 7: Comparison of probability density curves of $\mathcal{N}(0, \frac{ms^2}{n})$ (ND) and $\tilde{s}X$ under different $m$.

is more similar to E2LSH. As shown in Table 10, $\epsilon$ and $\lambda$ decrease as $m$ increases. Take *Trevi* as an example, $\epsilon$ is equal to 0 and $\lambda$ is very tiny (0.0001-0.000729), manifesting the equivalence between $f_{\tilde{s}X}(t)$ and the PDF of $\mathcal{N}(0, \frac{ms^2}{n})$.

Table 10: $\epsilon$ and $\lambda$ for different $m$

| Datasets | | $m=15$ | | $m=30$ | | $m=45/\mathbf{60}$ | | $m=n$ | |
|---|---|---|---|---|---|---|---|---|---|
| | | $\epsilon$ | $\lambda$ | $\epsilon$ | $\lambda$ | $\epsilon$ | $\lambda$ | $\epsilon$ | $\lambda$ |
| Cifar | max | 0.567575 | 2.527575 | 0.287600 | 1.287600 | **0.114125** | **0.618225** | 0.000064 | 0.067664 |
| | mean | 0.102716 | 0.575372 | 0.027539 | 0.277239 | **0.001994** | **0.120123** | 0 | 0.014617 |
| | min | 0.019616 | 0.239671 | 0.001702 | 0.116014 | **0.000011** | **0.055001** | 0 | 0.006691 |
| Sun | max | 0.502360 | 2.218460 | 0.236626 | 1.084867 | **0.095099** | **0.546683** | 0.000006 | 0.051535 |
| | mean | 0.022783 | 0.255107 | 0.001849 | 0.118130 | **0.000018** | **0.058099** | 0 | 0.006724 |
| | min | 0 | 0.040401 | 0 | 0.020736 | **0** | **0.010000** | 0 | 0.001156 |
| Gist | max | 0.633640 | 2.853740 | 0.274502 | 1.234902 | **0.129940** | **0.677540** | 0 | 0.032400 |
| | mean | 0.042122 | 0.340238 | 0.005183 | 0.152639 | **0.000133** | **0.074662** | 0 | 0.004624 |
| | min | 0.000064 | 0.067664 | 0 | 0.038025 | **0** | **0.016926** | 0 | 0.001089 |
| Trevi | max | 0.013420 | 0.207020 | 0.001105 | 0.106081 | **0.000003** | **0.049287** | 0 | 0.000729 |
| | mean | 0.000011 | 0.055236 | 0 | 0.029241 | **0** | **0.013924** | 0 | 0.000196 |
| | min | 0 | 0.015876 | 0 | 0.008100 | **0** | **0.003969** | 0 | 0.000100 |
| Audio | max | 0.268000 | 1.208900 | 0.099004 | 0.561404 | 0.043520 | 0.346020 | 0.000033 | 0.062533 |
| | mean | 0.033344 | 0.303017 | 0.003637 | 0.138767 | 0.000480 | 0.091021 | 0 | 0.020967 |
| | min | 0.000022 | 0.059705 | 0 | 0.030765 | 0 | 0.021874 | 0 | 0.005329 |
| Notre | max | 0.341109 | 1.507509 | 0.143598 | 0.728823 | 0.074226 | 0.467355 | 0.004017 | 0.142401 |
| | mean | 0.022783 | 0.255107 | 0.001902 | 0.118866 | 0.000172 | 0.077456 | 0 | 0.027225 |
| | min | 0.000101 | 0.071925 | 0 | 0.036481 | 0 | 0.023716 | 0 | 0.008281 |
| Glove | max | 0.261530 | 1.183130 | 0.094131 | 0.543031 | 0.040065 | 0.331665 | 0.002362 | 0.124862 |
| | mean | 0.003190 | 0.134234 | 0.000047 | 0.065072 | 0.000001 | 0.043265 | 0 | 0.019853 |
| | min | 0.000025 | 0.060541 | 0 | 0.029929 | 0 | 0.020335 | 0 | 0.009409 |
| Sift | max | 0.294194 | 1.314294 | 0.098513 | 0.559554 | 0.057014 | 0.400410 | 0.001296 | 0.109537 |
| | mean | 0.054265 | 0.389506 | 0.007185 | 0.167986 | 0.001509 | 0.113065 | 0 | 0.036864 |
| | min | 0.003755 | 0.139916 | 0.000107 | 0.072468 | 0.000001 | 0.045370 | 0 | 0.016129 |
| Deep | max | 0.036744 | 0.317644 | 0.003841 | 0.140741 | 0.000463 | 0.090463 | 0 | 0.014400 |
| | mean | 0.003047 | 0.132719 | 0.000044 | 0.064560 | 0.000001 | 0.043306 | 0 | 0.007569 |
| | min | 0.000199 | 0.079160 | 0 | 0.039204 | 0 | 0.026374 | 0 | 0.004638 |
| Random | max | 0.077344 | 0.479300 | 0.015195 | 0.216796 | 0.003505 | 0.137461 | 0.000041 | 0.064050 |
| | mean | 0.002824 | 0.130273 | 0.000038 | 0.063542 | 0.000001 | 0.042437 | 0 | 0.019044 |
| | min | 0.000024 | 0.060049 | 0 | 0.029929 | 0. | 0.019881 | 0 | 0.008649 |
| Ukbench | max | 0.692955 | 3.157855 | 0.361581 | 1.593681 | 0.217238 | 1.009338 | 0.039727 | 0.330248 |
| | mean | 0.139183 | 0.712232 | 0.034502 | 0.308031 | 0.012672 | 0.202768 | 0.000056 | 0.066620 |
| | min | 0.002895 | 0.131059 | 0.000041 | 0.064050 | 0.000001 | 0.042437 | 0 | 0.015376 |
| ImageNet | max | 0.466568 | 2.054168 | 0.217238 | 1.009338 | 0.119323 | 0.637723 | 0.004773 | 0.149173 |
| | mean | 0.019125 | 0.237214 | 0.001336 | 0.110236 | 0.000107 | 0.072468 | 0 | 0.022201 |
| | min | 0 | 0.033489 | 0 | 0.016641 | 0 | 0.011025 | 0 | 0.003481 |

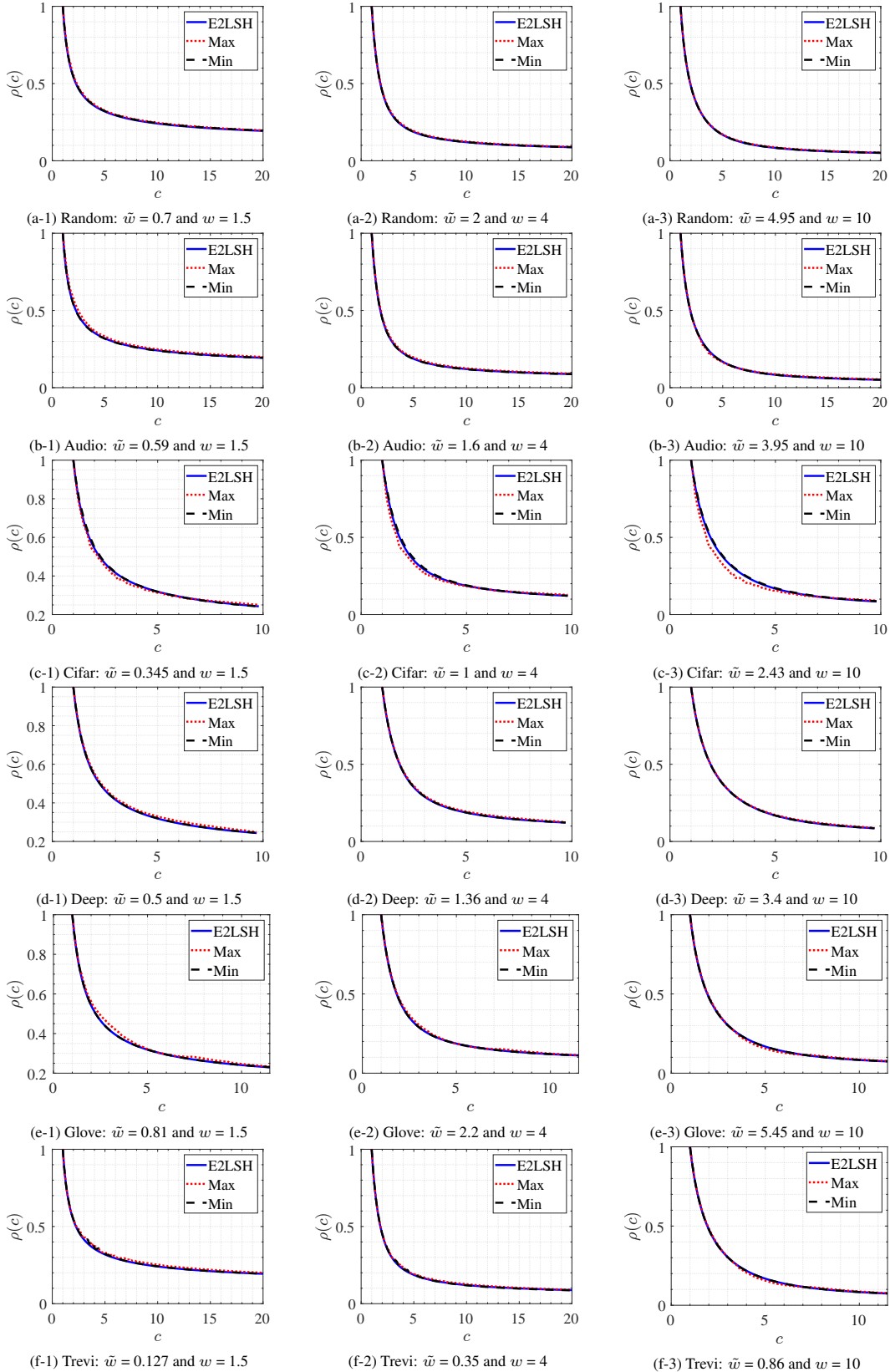

Figure 8: $\rho$ curves under different bucket widths over datasets *Random*, *Audio* and *Cifar*, *Deep*, *Glove* and *Trevi*.

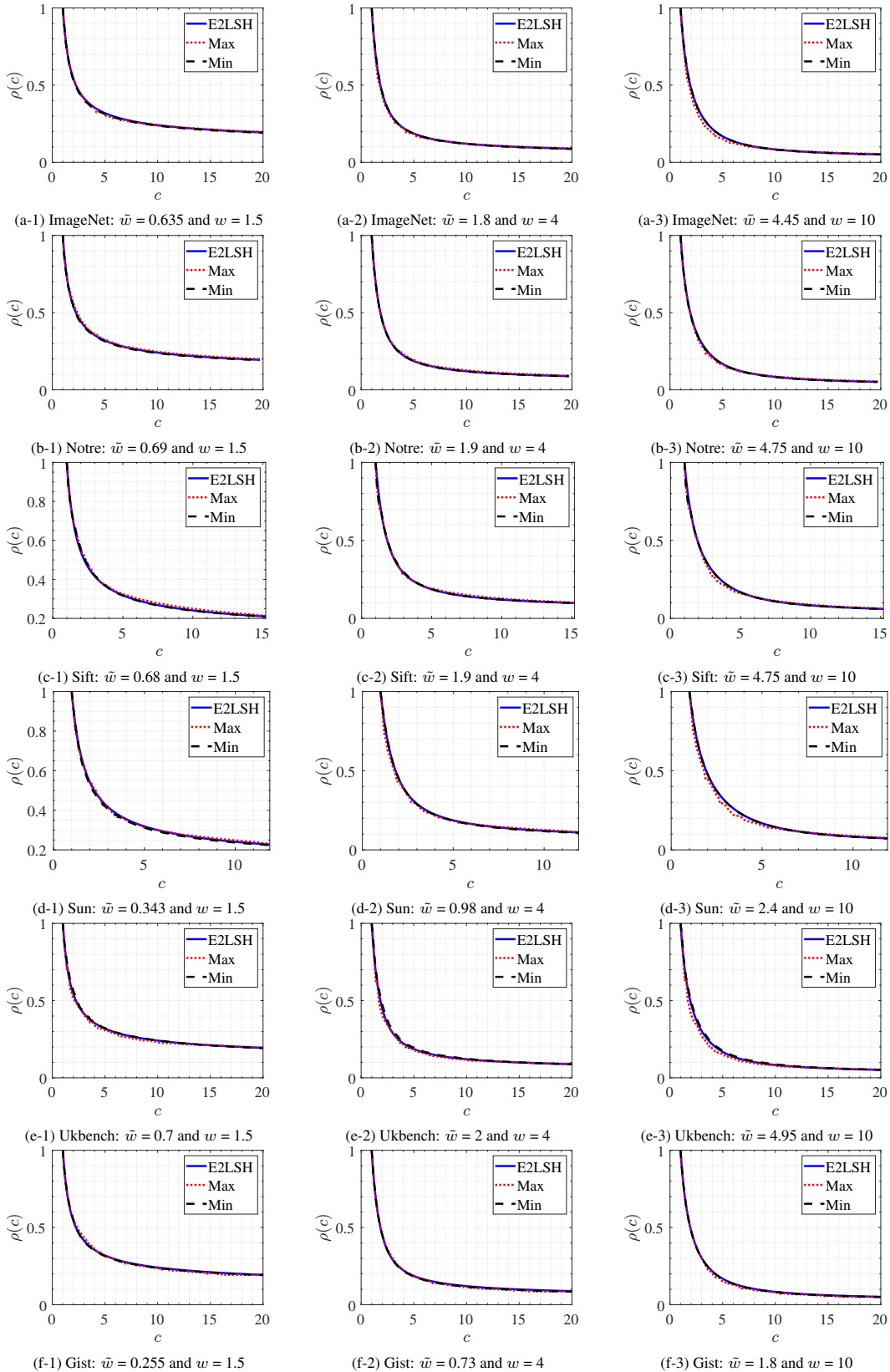

Figure 9: $\rho$ curves under different bucket widths over datasets *ImageNet*, *Notre*, *Sift*, *Sun*, *Ukbench* and *Gist*.

