# OpenReview forum: "Simple Yet Efficient Locality Sensitive Hashing with Theoretical Guarantee"
_ICLR.cc/2025/Conference — Submitted to ICLR 2025_

### Official Review · Reviewer_yqFQ · 2024-10-19

**Soundness:** 2
**Presentation:** 2
**Contribution:** 2
**Rating:** 5
**Confidence:** 4

**Summary:**

This paper proposes a new LSH method, named FastLSH, which aims to accelerate the indexing process of traditional LSH. The authors provide theoretical analysis to argue that their approach retains the fundamental LSH property: that the closer two points are, the higher the probability they will collide in the same hash bucket. The empirical contribution is presented through three groups of experiments, where FastLSH is applied in outlier detection, specialized neural network training (SLIDE) using LSH, and nearest neighbor search.

**Strengths:**

This paper presents a new approach, FastLSH, which introduces  a new improvement to the indexing process of canonical LSH. The idea of selectively sampling dimensions to accelerate indexing is well-motivated, and the authors attempt to show they retain the core LSH property through both theoretical analysis and empirical validation. They do extensive experiments. The paper is easy to understand.

**Weaknesses:**

**W1. Weak Justification of the Research Problem’s Value**

The paper does not sufficiently establish the practical importance of accelerating the indexing process for LSH. Hashing-based methods are already among the fastest algorithms for indexing when compared to quantization-, tree-, and graph-based methods. Among these, LSH is known for fast indexing. From the three applications discussed (outlier detection, neural network training, and nearest neighbor search), it is not evident that further improving LSH indexing speed is a critical need.

For example, in the nearest neighbor search application, the authors report that indexing the GIST1M dataset (960-dimension, 1 million points) takes under 30 seconds—which is already sufficiently fast for most applications. The paper’s focus on indexing overlooks a more pressing issue: search efficiency, where LSH typically performs poorly compared to recent graph-based methods such as HNSW [1]. Improving search efficiency would address a more significant problem, which is why little effort in the literature is devoted to accelerating LSH indexing. While the SLIDE framework may benefit from faster indexing due to frequent re-indexing, it is a specialized case, and more examples are needed to justify the broader value of this work.

**W2. Theoretical Flaws and Insufficient Justification**

The theoretical analysis contains several potential flaws and requires further rigorous justifications to support the authors' claims:

**W2.1 Central Limit Theorem (CLT) Application:**

In Lemma 4.1, the authors apply the CLT, but the conditions for its use are not fully satisfied. CLT assumes i.i.d. samples, yet the proposed method ensures only that the selected dimensions are independently sampled. The elements within each dimension may not be i.i.d. since the authors assume only finite mean and variance for the distribution of  $(u_i-v_i)^2$. Since the underlying distribution is unknown, more justification is needed to ensure that the CLT applies. This step is crucial since it forms the foundation for the entire theoretical framework.

**W2.2 Asymptotic Convergence of Characteristic Functions:**

In Theorem 4.6, the authors aim to demonstrate that the ratio of the characteristic functions of the original and transformed distributions converges to 1 as $m\rightarrow \infty $. However, the convergence depends on the behavior of the input to the characteristic function, x. The authors must show that the ratio of the characteristic functions converge for all values of x. The claim that $x^2\leq O(m^{-1})$ is “obvious” is problematic, as no rigorous justification is provided. This is critical since the convergence ratio will diverge for non-zero x if this condition does not hold.

**W2.3 Practicality of the Asymptotic Results:**

Even if Theorem 4.6 holds, the requirement that $m\rightarrow \infty $ raises practical concerns. Since the authors propose using fewer dimensions (m < n), they must demonstrate that the asymptotic results still hold in practice. Specifically, the authors should quantify how far the transformed distribution deviates from the target distribution under finite m and provide a lower bound for the distribution distance under a suitable metric. Section 4.3 contains only heuristic arguments, making it unclear to what extent the LSH property is preserved after dimension sampling. A more rigorous analysis is required to confirm that the proposed method retains the LSH property, or at least an approximate version of it.

**W3. Weaknesses in Experimental Design and Results**

Several flaws in the experimental design limit the contribution of this paper:

**W3.1 Application Scenarios Are Not Well-Aligned with Claims:**

The authors argue that their method is beneficial for scenarios requiring frequent re-indexing. However, the experiments do not reflect such settings. For example, no experiments are conducted on streaming data, which would be a more relevant use case. Moreover, the authors should compare their method to state-of-the-art approaches for high-dimensional data streams, such as [2][3].

**W3.2 Incomplete Use of Standard Datasets:**

The authors use well-known datasets such as SIFT, Glove, and GIST, but they do not utilize all queries in these datasets. For instance, the SIFT dataset contains 10,000 queries, yet only 200 were used in the experiments. This is unusual, and the paper provides no justification for this choice. The authors should explain how the subset was selected and whether this affects the search performance.

**W3.3 Diverse Speedup in Outlier Detection Task:**

The reported speedup of FastLSH over baseline methods varies significantly across datasets, especially in the outlier detection task. This raises concerns about whether the proposed method introduces distortions in the Hamming distances or requires dataset-specific hyper-parameter tuning. Either issue would limit the generality of the method and should be thoroughly investigated and reported.

**Questions:**

S1: Strengthen the theoretical analysis and address the problems in W2
S2: Add suitable experiments and give proper analysis to demonstrate the strengths of this paper to reach ICLR standard.
S3. Justify the research value of this problem with broad use cases.

---

> ### Author Response · Authors · 2024-11-15
> **Comment to Reviewer yqFQ**
>
> Thanks for your feedback. Next we will address the main points you raised.
>
> ## Response to W1
> Our work focuses on using LSH-based method to accelerate end-to-end index construction, supporting machine learning tasks such as anomaly detection, neural network training, and ANN search. It is worth noting that FastLSH and ACHash do not necessarily reduce query time; their primary purpose is to speed up the hash evaluations. More importantly, LSH can be applied to other tasks beyond ANN search, such as outlier detection and neural network training. For these tasks, only $k \times L$ hash tables need to be built, and distance computations are not required. Additionally, these tasks might require frequent creation or updating of hash tables, such that the execution time is mainly dominated by the hashing cost, then a notable characteristic is that index construction time is more important than query processing. In such applications, FastLSH significantly reduces the hashing cost, as shown in Figure 3, Table 1, Table 6 and Table 7 in Appendix C.3. Although many graph-based methods (e.g., HNSW [1]) achieve higher query accuracy in ANN search, they are not suitable for accelerating end-to-end index construction in these applications. Research has shown that when HNSW is used to speed up neural network training, its time consumption is **23** times higher than using LSH, and HNSW does not have guarantees for search performance [2].
>
> Regardless of whether it is the SLIDE framework or other frameworks (ACE [3], et al), if LSH is used for hashing involving inner product computations, FastLSH can integrate well into these frameworks, e.g., [4-6]. This is because, in the case of $m < n$, Lemma 4.8 and Fact 4.9 rigorously prove that FastLSH and E2LSH are very similar, by comparing the first four moments of the two distributions, despite the presence of data-dependent parameters $\epsilon$ and $\lambda$ introduced by $\sigma$. Additionally, the empirical results in Table 9 in Appendix show that as $m$ increases, the influence of these parameters $\epsilon$ and $\lambda$ becomes very small, further indicating that FastLSH and E2LSH are very close.
>
> It should be noted that the end-to-end index construction time for Gist1M is not approximately 30 seconds; we have scaled the results for better display, as shown in Figures 4(d) and 6(d). In fact, even when using FastLSH, with a sampled dimension of $m=30$ and under our $k \times L$ settings, the index construction time takes at least 200 seconds. In contrast, when using 960 dimensions with E2LSH, the time consumption is even higher, with end-to-end index construction taking over a thousand seconds.
>
> [1] Cong Fu, Chao Xiang, Changxu Wang, and Deng Cai. Fast approximate nearest neighbor search with the navigating spreading-out graph. arXiv preprint arXiv:1707.00143, 2017.
>
> [2] Chen, Beidi, et al. Mongoose: A learnable lsh framework for efficient neural network training. In Proceedings of International Conference on Learning Representations. 2020.
>
> [3] Luo, Chen, and Anshumali Shrivastava. Arrays of (locality-sensitive) count estimators (ace) anomaly detection on the edge. In Proceedings of the 2018 World Wide Web Conference. 2018.
>
> [4] Chen, Beidi, et al. Mongoose: A learnable lsh framework for efficient neural network training. In Proceedings of International Conference on Learning Representations. 2020.
>
> [5] Kitaev, Nikita, Łukasz Kaiser, and Anselm Levskaya. Reformer: The efficient transformer. arXiv preprint arXiv:2001.04451 (2020).
>
> [6] Rabbani, Tahseen, Marco Bornstein, and Furong Huang. Large-Scale Distributed Learning via Private On-Device LSH. In Proceedings of Advances in Neural Information Processing Systems 36 (2024).
>
> ## Response to W2.1
>
> For given vector pair $(\mathbf{v},\mathbf{u})$, let $s = || \mathbf{v}-\mathbf{u} ||$. For our purpose, assume the collection of $n$ entries $(v_{i}-u_{i})^{2}$ $(i=1,2,\ldots,n)$ is a population, which follows an unknown distribution with mean $\mu =( {\textstyle \sum_{i=1}^{n}}(v_{i}-u_{i})^{2}) /n$ and variance $\sigma^{2}=({\textstyle \sum_{i=1}^{n}}((v_{i}-u_{i})^{2}-\mu)^{2})/ n $. It is obvious that, for any given pair of vectors of finite dimension $n$, the 2-norm of the difference across each dimension MUST has a finite mean $\mu$ and finite variance $\sigma^2$. Here, each entry $(v_{i}-u_{i})^{2}$ for $(i=1,2,\ldots,n)$ is independently sampled from an unknown distribution. Therefore, we can use the CLT [1] to derive Lemma 4.1.
>
> [1] https://en.wikipedia.org/wiki/Central_limit_theorem

---

> > ### Author Response · Authors · 2024-11-15
> > **Continued Comment to Reviewer yqFQ**
> >
> > ## Response to W2.2
> > The validity of Theorem 4.6 is conditional, i.e., $|x|\leq O(m^{-1/2})$. Theorem 4.6 implies that $\varphi_{\tilde{s}X}(x)$ is asymptotically identical to $\exp(-\frac{ms^{2}x^{2}}{2n})$ within interval $[-\mathcal{K}\sqrt{\frac{n}{ms^{2}}}, +\mathcal{K}\sqrt{\frac{n}{ms^{2}}}]$ (i.e., $|x|\leq O(m^{-1/2})$), that is, 2$\mathcal{K}$ ``standard deviations", where $\mathcal{K}$ is an arbitrarily large constant. The goal is that when $\mathcal{K}$ is sufficiently large, the effective components of the distributions of FastLSH and E2LSH are essentially identical. What we are proving here is the equivalence under the condition of $x$, not the equivalence for all values of $x$. Since we introduced the data-dependent quantity $\sigma$ in Lemma 4.1, which makes FastLSH a data-dependent LSH, it is intractable for FastLSH to achieve the same level of perfect theoretical analysis as traditional data-independent LSH (E2LSH). For this issue, we will add a clearer description to specify the conditions under which Theorem 4.6 holds.
> >
> >
> > ## Response to W2.3
> > In practical scenarios, $m$ is often limited. We study the relation between $f_{\tilde{s}X}(t)$ and the PDF of $\mathcal{N}(0,\frac{ms^{2}}{n})$ when $m$ is relatively small ($m < n$). Particularly, we derive the first four moments of $\tilde{s}X$ and $\mathcal{N}(0,\frac{ms^{2}}{n})$, and analyze how $m$ and $\sigma$ affect their similarity. While in general the first four moments, or even the whole moment sequence may not determine a distribution [1], practitioners find that distributions near the normal can be decided very well given the first four moments [2,3]. To this end, we derive Lemma 4.8 rigorously, which is not merely a heuristic argument.
> >
> > Then we quantitatively analyzed the difference and manifests that the gap between FastLSH and LSH can be captured by parameters $\epsilon$ and $\lambda$ (Lemma 4.8 and Fact 4.9). In short, greater $m$ is, then smaller $\epsilon$ and $\lambda$ will be. Thus, by choosing appropriate $m$ ($m$ is set 30 in comparison with E2LSH), $\epsilon$ and $\lambda$ are small enough to reduce the impact of the variance in the squared distances of coordinates to a negligible level, which makes FastLSH and LSH are practically equivalent. Table 10 in Appendix C.8 illustrates the empirical evidences on how $m$ affects $\epsilon$ and $\lambda$ across 12 datasets we are experimented with. Furthermore, Figure 8 and Figure 9 in Appendix C.7 illustrate a comparison of the $\rho$ curves (an important measure of the LSH property) for E2LSH and FastLSH. These figures show that their $\rho$ curves match well across different datasets, verifying that FastLSH and E2LSH have the same LSH performance. As stated in W2.2, FastLSH is a data-dependent LSH, due to the introduced data-dependent number $\sigma$, A more rigorous theoretical analysis is intractable. To the best of our knowledge, there is currently no data-dependent LSH that provides LSH property similar to data-independent LSH (E2LSH).
> >
> > [1] Lin, G. D. Recent developments on the moment problem. Journal of Statistical Distributions and Applications, 4 (1):5, 2017.
> >
> > [2] Leslie, D. Determination of parameters in the johnson system of probability distributions. Biometrika, 46(1/2): 229–231, 1959.
> >
> > [3] Ramberg, J. S., Dudewicz, E. J., Tadikamalla, P. R., and Mykytka, E. F. A probability distribution and its uses in f itting data. Technometrics, 21(2):201–214, 1979.
> >
> >
> > ## Response to W3.1
> > The SLIDE framework effectively demonstrates that FastLSH provides significant acceleration for frequently constructing hash tables while also improving query accuracy, even though it does not handle streaming data. Lemma 4.8 and Fact 4.9 provide an analysis of the distributional difference between FastLSH and E2LSH. By adjusting the size of $m<n$, FastLSH can asymptotically become equivalent to E2LSH. If E2LSH effectively handles streaming data, we believe FastLSH is also applicable. Regarding the references [2, 3] you mentioned, if time permits, we will add comparative experiments to further show the broad applicability of FastLSH. However, could you clarify which two papers you are referring to with [2, 3]?

---

> > > ### Author Response · Authors · 2024-11-15
> > > **Continued Comment to Reviewer yqFQ**
> > >
> > > ## Response to W3.2
> > > For ANN search tasks, many papers [1-4] randomly select a subset from the given query set, such as 50, 100, or 200. We have also adopted this approach.
> > >
> > > [1] Huang, Qiang, et al. Query-aware locality-sensitive hashing for approximate nearest neighbor search. In Proceedings of the VLDB Endowment 9.1 (2015): 1-12.
> > >
> > > [2] Sun, Yifang, et al. SRS: solving c-approximate nearest neighbor queries in high dimensional euclidean space with a tiny index. In Proceedings of the VLDB Endowment (2014).
> > >
> > > [3] Lei, Yifan, et al. Locality-sensitive hashing scheme based on longest circular co-substring. In Proceedings of the 2020 ACM SIGMOD International Conference on Management of Data. 2020.
> > >
> > > [4] Tian, Yao, Xi Zhao, and Xiaofang Zhou. DB-LSH 2.0: Locality-sensitive hashing with query-based dynamic bucketing. IEEE Transactions on Knowledge and Data Engineering (2023).
> > >
> > >
> > > ## Response to W3.3
> > > The acceleration of FastLSH does not require any hyperparameter tuning across datasets and is solely dependent on the chosen sampling dimension $m$. Since we report the total execution time for the anomaly detection task, which includes both index construction and query time, the execution time appears to vary significantly due to different dataset sizes and choices of $m$, as shown in Table 4 and Parameter Settings in Appendix C.1. Furthermore, the results in the anomaly detection and neural network training tasks, as shown in Table 1, Table 2 and Table 3 and Figure 2, demonstrate that FastLSH does not introduce distortions in the Hamming distances.

---

> > > > ### Comment · Reviewer_yqFQ · 2024-11-15
> > > > **Response to Author Feedbacks**
> > > >
> > > > **Response w.r.t. W2.1 Feedback**
> > > >
> > > > As shown in CLT, the application condition assumes a sequential **i.i.d. random samples**. Though the dimensions are independently selected, whether the samples, i.e. the samples $(v_i-u_i)^2$, are i.i.d is questionable. IMHO, the i.i.d. assumption may hold if $v_i$ across different dimensions and samples should also be drawn i.i.d. from a distribution.
> > > >
> > > > **Response w.r.t. W2.3 Feedback**
> > > >
> > > > Given that more rigorous theoretical guarantee is not given. The term "with theoretical guarantee" is to some extent an overclaim. With the problem in W2.2 not properly addressed, this reduces the theoretical contribution of this paper.
> > > >
> > > > **Response w.r.t. W3.1 Feedback**
> > > >
> > > > Sorry for missing the references, as below:
> > > > [1] Malkov, Yu A., et al. "Efficient and robust approximate nearest neighbor search using hierarchical navigable small world graphs."
> > > > [2] Yang, Chengcheng, et al. "Efficient locality-sensitive hashing over high-dimensional data streams."
> > > > [3] Wang, Hao, et al. "Efficient locality-sensitive hashing over high-dimensional streaming data."
> > > >
> > > > **Response w.r.t. W3.2 Feedback**
> > > > Though the references are given, majority works in this literature follows the official split of these datasets. Given the fact that hashing based methods are quite fast, it is unnecessary to select a subset of queries. By doing so, it is more difficult to position the contribution of this paper in a wider range of literature.
> > > >
> > > > **Summary**
> > > >
> > > > Given the feedback from the authors, the problems within the theoretical framework are not well addressed, especially the authors admit the flaws in Theorem 4.6, which is a key part. The term "with theoretical guarantee of LSH properties" is somewhat overclaimed.
> > > >
> > > > Meanwhile, the empirical assessments are not sufficient to show the value in terms of wide application of the proposed methods. For example, given the authors have provided more works using LSH in model training, they should provide more experiments rather than just for SLIDE.
> > > >
> > > > IMHO, to well position the contribution of a simple methodology, one can either provide rigorous theoretical guarantees, or conduct extensive and comprehensive experiments to demonstrate their wide usability. Given the current status of this paper and the feedback, I may not change the score.

---

### Official Review · Reviewer_sEFP · 2024-11-03

**Soundness:** 3
**Presentation:** 3
**Contribution:** 2
**Rating:** 5
**Confidence:** 3

**Summary:**

This paper introduces FastLSH, a novel locality-sensitive hashing scheme that combines random sampling and random projection to reduce the hashing complexity from $O(n)$ to $O(m)$, where $m$ is the number of samplings and $m<n$.
FastLSH is claimed to preserve the LSH properties, i.e., the collision probability can be calculated like that in E2LSH.
The faster hash computations in FastLSH make it well-suited for tasks like anomaly detection, neural network training, and nearest neighbor search.

**Strengths:**

1. FastLSH is simple, it can be easily implemented and can be seamlessly integrated into existing LSH-based applications.

2. The paper provides rigorous theoretical proofs that FastLSH retains the desirable LSH properties.

**Weaknesses:**

1. The theoretical guarantees hold only when the number of sampled dimensions, $m$, approaches infinity.
In real-world applications, it is possible to construct dense data sets on which FastLSH may fail. For example, consider a data set in which a large proportion of dimensions are the same, with only a few being non-trivial.
In such cases, FastLSH is likely to miss the non-trivial dimensions during the sampling process and end up hashing all data points into the same bucket. This raises concerns about potential theoretical flaw in FastLSH.
Therefore, it would be beneficial for the authors to demonstrate the effectiveness of FastLSH on relatively rare and challenging scenarios, as exemplified above.

2. While FastLSH reduces the cost of hash function computations and index construction time, it does not speed up the query process itself. This limits its broader impact on applications where query speed is critical.

3. The paper does not sufficiently explore the effect of varying the parameter $m$, the number of sampled dimensions, on both the efficiency and accuracy of FastLSH.
Since $m$ plays a crucial role in balancing computational savings with hashing accuracy, understanding its impact across a range of values is essential.
Without a thorough parameter study, it remains unclear how to optimally set $m$ for different datasets or applications.

**Questions:**

1. How does FastLSH handle scenarios where only a few dimensions carry critical information, while others are redundant? Could you provide experiments on such challenging data sets? (W1)

2. Moreover, is there a mechanism in FastLSH to adaptively select informative dimensions during sampling? There are existing methods that adaptively sample the dimensions based on their informativeness, with a non-uniform distribution. (W1)

3. Could you include a parameter study showing how different values of $m$ affect performance across various data sets? (W3)

4. Furthermore, could you provide guidelines or heuristics on how to choose $m$ for a given application? (W3)

---

> ### Author Response · Authors · 2024-11-15
> **Comment to Reviewer sEFP**
>
> Thanks for your feedback. Next we will address the main points you raised.
>
> ## Response to (W1)
> We are already aware of the issue you have raised. FastLSH can handle sparse data, as discussed in Appendix C.6. We use the MNIST dataset to validate this claim, and the results are shown in Table 9 in Appendix, where MNIST is a sparse dataset with around 2.6% non-zero elements. As you have focused on, for very sparse vectors, we can use the Hadamard transform to make the data dense. This makes FastLSH and AChash similar, so FastLSH can be considered a generalized version of AChash. In many practical applications, data is often dense, in which case FastLSH can be used directly. For sparse vectors, we apply the Hadamard transform for data preprocessing. This enhances the adaptability of FastLSH compared to AChash, enabling faster index construction and better query performance.
>
>
> ## Response to (W2)
> It is worth noting that FastLSH and ACHash [1] do not necessarily reduce query time; their primary purpose is to speed up the hash evaluations. More importantly, LSH can be applied to other tasks beyond ANN search, such as outlier detection and neural network training. For these tasks, only $k \times L$ hash tables need to be built, and distance computations are not required. Additionally, these tasks might require frequent creation or updating of hash tables, such that the execution time is mainly dominated by the hashing cost. In such applications, FastLSH significantly reduces the hashing cost, as shown in Figure 3, Table 1, Table 6 and Table 7 in Appendix C.3.
>
> [1] Dasgupta, Anirban, Ravi Kumar, and Tamás Sarlós. Fast locality-sensitive hashing. In Proceedings of the 17th ACM SIGKDD international conference on Knowledge discovery and data mining. 2011.
>
> ## Response to (W3)
> We make a comprehensive analysis of $m$. The first result is illustrated in Theorem 4.6 and Corollary 4.7, where the asymptotic analysis of FastLSH indicates the equivalence between FastLSH and the classic LSH as $m$ approaches infinity. This analysis is aimed to show the relationship between our proposal and LSH from a theoretical perspective.
>
> In practice, however, $m$ is expected to be a small number (less than $n$) to make FastLSH useful. In this case, the difference between FastLSH and LSH is controlled by $m$ and the variance in the squared distances of coordinates of a pair of data items, which is data-dependent and has nothing to do with $n$.
>
> Our second main result (Lemma 4.8 and Fact 4.9) quantitatively analyzed the difference and manifests that the gap between FastLSH and LSH can be captured by parameters $\epsilon$ and $\lambda$. In short, greater $m$ is, then smaller $\epsilon$ and $\lambda$ will be. Thus, by choosing appropriate $m$ ($m$ is set 30 in comparison with E2LSH), $\epsilon$ and $\lambda$ are small enough to reduce the impact of the variance in the squared distances of coordinates to a negligible level, which makes FastLSH and LSH are practically equivalent. Table 10 in Appendix C.8 illustrates the empirical evidences on how $m$ affects $\epsilon$ and $\lambda$ across 12 datasets we are experimented with.
>
> In practice, the value of $m$ can be set in the range of $[30, \frac{n}{2}]$. FastLSH provides the default $m$ settings, as shown in Parameter Settings in Appendix C.1, C.2, C.3 and C.4.

---

> > ### Comment · Reviewer_sEFP · 2024-11-24
> >
> > Thank you for your detailed response! It addressed some of my concerns, and I have some follow-up questions and request for clarification:
> >
> > **Regarding (W1):**
> >
> > I appreciate your explanation of applying the Hadamard transformation as a solution. While I agree that it can help mitigate the issue, I still have the following concerns:
> >
> > (1) By applying the Hadamard transformation, the dataset is converted into a dense one. It remains unclear whether FastLSH is still effective when handling sparse or maliciously designed datasets.
> >
> > (2) This solution makes FastLSH appear similar to ACHash. Could you elaborate more on the advantages of FastLSH in this context?
> >
> > **Regarding (w2):**
> >
> > I mostly agree with your response here, but I have a few follow-up questions:
> >
> > (1) In Figure 3, there are two rows of plots with the same x-axis and y-axis. Why not combine these rows for a clearer and more concise presentation? Are the two rows using different parameter settings, or is there another reason for separating these methods?
> >
> > (2) In the results presented in Figure 3, the speed-up in indexing time appears to be significant only for the Trevi dataset. Could you explain why FastLSH performs better on Trevi compared to the other datasets? Is its performance dataset-dependent, and what conditions of a dataset favor its performance?
> >
> > (3) While the speed-up achieved by FastLSH on the Musk dataset (Table 1) is significant, its improvement on the Statlog Shuttle dataset (Table 3) seems relatively modest. Similar to question (2), could you provide additional explanations or experimental results to clarify whether FastLSH consistently achieves significant speed-ups across different datasets?
> >
> > **Regarding (W3):**
> >
> > I appreciate that the response highlighted experiments with various small values of $m$. However, my initial question aimed to understand why, the infinite $m$ in theory can often be replaced by a small $m$ while still maintaining correctness in practice. Since this concern wasn’t fully addressed, could you provide more theoretical analysis (or empirical insights) to clarify this point?

---

### Official Review · Reviewer_XLCc · 2024-11-04

**Soundness:** 2
**Presentation:** 3
**Contribution:** 2
**Rating:** 5
**Confidence:** 4

**Summary:**

This paper aims at the efficiency of LSH methods while not harming its effectiveness. It reduces the cost of computing hashing functions by random sampling. The authors verify the effectiveness of their methods.

**Strengths:**

S1. The method seems sound.

S2. This paper studies important problems.

S3. It is well written.

**Weaknesses:**

W1. The experiments focus on LSH based methods for ANNS, outlier detection. However, there are other methods such as proximity graphs for ANNS and OD. Besides, LSH based methods are not the SOTA for both of them. Even though LSH methods are enhanced, it does not really make a progress to ANNS and OD.

**Questions:**

Q1. I would see more experiments to demonstrate the the method in this paper outperform the SOTA method for ANNS and outlier detection.

---

> ### Author Response · Authors · 2024-11-15
> **Comment to Reviewer XLCc**
>
> Thanks for your feedback. Next we will address the main points you raised.
> ## Response to Weaknesses
> Our work focuses on using LSH-based method to accelerate end-to-end index construction, supporting machine learning tasks such as anomaly detection, neural network training, and ANN search. For LSH-based applications (e.g., the ACE method for anomaly detection [1]), its most significant advantages lie in low memory usage (required less than **4MB** memory) and fast query processing (reached up to **150x**), while maintaining comparable query accuracy to other advanced methods. We have chosen ACE as a SOTA method. Although many graph-based methods (e.g., HNSW [2]) achieve higher query accuracy in ANN search, they are not suitable for accelerating end-to-end index construction in LSH-based applications. Research has shown that when HNSW is used to speed up neural network training, its time consumption is **23** times higher than using LSH, and HNSW does not have guarantees for search performance [3]. For these LSH-based applications, a notable characteristic is that index construction time is more important than query processing, and the execution time is mainly dominated by the hashing cost, so that FastLSH significantly reduces the hashing cost, as shown in Figure 2, Table 1, Table 2 and Table 3. Furthermore, LSH is a fundamental component for high-dimensional ANN search, and we selected E2LSH [4] and MPLSH [5] as baselines because they are commonly used in practice. Our goal is to verify why FastLSH can improve query accuracy in anomaly detection and neural network training tasks, as shown in Figure 2, Table 1, Table 2 and Table 3. This is because FastLSH not only achieves comparable query accuracy to E2LSH and MPLSH but also significantly speeds up end-to-end index construction, as shown in Figure 3, Figure 4, Figure 6 and Table 9 in Appendix.
>
> [1] Luo, Chen, and Anshumali Shrivastava. Arrays of (locality-sensitive) count estimators (ace) anomaly detection on the edge. In Proceedings of the 2018 World Wide Web Conference. 2018.
>
> [2] Cong Fu, Chao Xiang, Changxu Wang, and Deng Cai. Fast approximate nearest neighbor search with the navigating spreading-out graph. arXiv preprint arXiv:1707.00143, 2017.
>
> [3] Chen, Beidi, et al. Mongoose: A learnable lsh framework for efficient neural network training. In Proceedings of International Conference on Learning Representations. 2020.
>
> [4] Datar, M., Immorlica, N., Indyk, P., and Mirrokni, V. S. Locality-sensitive hashing scheme based on p-stable distributions. In Proceedings of the twentieth annual symposium on Computational geometry, pp. 253–262, 2004.
>
> [5] Qin Lv, William Josephson, Zhe Wang, Moses Charikar, and Kai Li. Multi-probe lsh: efficient indexing for high-dimensional similarity search. In Proceedings of the 33rd international confer ence on Very large data bases, pp. 950–961, 2007.

---

### Official Review · Reviewer_ze7N · 2024-11-09

**Soundness:** 1
**Presentation:** 2
**Contribution:** 2
**Rating:** 3
**Confidence:** 4

**Summary:**

The paper focusses on making locality sensitive hashing (LSH) faster under the \ell_2 metric. The standard LSH scheme involves taking an inner product of the query with a random vector and bucketing the query according to the obtained value. The paper instead proposes to speed up this operation by first subsampling m coordinates of the vector and computing the inner product with the corresponding subsampled vector. It is shown that as m tends to infinity, the probability of collision under the proposed scheme is same as the standard LSH. The paper also shows the superior performance of the proposed scheme empirically.

**Strengths:**

Locality sensitive hashing is used widely, so any effort in speeding it up is welcome as it can have huge practical significance.

**Weaknesses:**

Dasgupta et. al. [1] came up with a two-step proposal to speed up the standard LSH using fast Johnson–Lindenstrauss transform. The LSH scheme proposed in this paper essentially removes the first step. However, this step is crucial especially when the dataset consists of sparse vectors. Thus, this paper seems to rediscover some of the ideas already present in [1], while missing the crucial ingredients.

More details:
The hash function proposed in [1] consists of two steps: (i) First multiply the query vector by a diagonal matrix with diagonal entries chosen to be 1 or -1 equiprobably. Then hit the vector obtained by a Hadamard matrix. (ii) Subsample roughly m coordinates of the resulting vector uniformly at random (without replacement), take the inner product of the resulting subsampled vector with a random gaussian vector and finally bucket the query according to the obtained value (sub sampling is actually done by choosing each coordinate with some fixed probability q = m/d). The first step is crucial when the vectors involved are sparse. In that case, most of the contribution to the \ell_2 distance comes from very few non-zero coordinates. Therefore, for the subsampling to be effective, m will need to be very high, defeating the main purpose. The first step applies a norm-preserving rotation to the vectors, with the desirable property that the vector so obtained is dense, that is, no entry is too large with high probability.

The scheme proposed in this paper essentially applies the second step but where the subsampling is done with replacement. However, not applying the first step means m will need to be very large for sparse vectors. That is why the paper could only show asymptotic equivalence (for m going to infinity) between the proposed scheme and the standard LSH. In contrast, Dasgupta et.al. prove that the collision probability of their proposed scheme is close to the standard LSH for m = O(log d).

[1] Dasgupta, Anirban, Ravi Kumar, and Tamás Sarlós. "Fast locality-sensitive hashing." Proceedings of the 17th ACM SIGKDD international conference on Knowledge discovery and data mining. 2011.

**Questions:**

Any clarification on points raised in the weaknesses section would be helpful.

---

> ### Author Response · Authors · 2024-11-15
> **Commnet to Reviewer ze7N**
>
> Thanks for your feedback. Next we will address the main points you raised.
>
> ## Difference Between FastLSH and ACHash
> To answer the question raised, we need to first make it clear what the provable LSH property is. A formal definition can be found in Def. 2.1, and a somewhat informal description is that the collision probability should decrease with the distance between the given pair of vectors. Only with the property, we can enjoy nice features that LSH-style algorithms deliver [1].
>
> For ACHash [2], unfortunately, this property does not hold because it can only offer a JL-transformation-style lower/upper bound on the collision probability. To be precise, suppose the distance between vectors $x_1$ and $y_1$ is $s_1$ and the distance between vectors $x_2$ and $y_2$ is $s_2$ and $s_1$ < $s_2$, it is impossible for ACHash to say the collision probability $p(s_1)$ > $p(s_2)$ because it only has information about the loose lower/upper bound on the collision probability. In a nutshell, the JL-transformation-style lower/upper bound cannot deliver the LSH property.
>
> In contrast to ACHash, FastLSH achieves this goal by deriving the exact collision probability DIRECTLY in Theorem 4.2, allowing one to calculate precisely the probability of collision for any pair of vectors with distance $s$. To deal with the new random variable $\tilde{s}X$, we overcome quite a lot technical difficulty (Lemma 4.4 and Appendix A.2) to derive the PDF of $\tilde{s}X$ in Eqn. (9). Figure 8 and Figure 9 in Appendix C.7 illustrate a comparison of the $\rho$ curves, an important measure of the LSH property, for E2LSH [1] and FastLSH. These figures show that their $\rho$ curves match well across different datasets, verifying that FastLSH and E2LSH have the same LSH performance. Note that it is impossible ACHash to plot such $\rho$ curves.
>
> Overall, AChash does not possess provable LSH property. Although FastLSH and AChash are similar in some aspects, the theoretical analysis methods of AChash are fundamentally different from those of FastLSH.
>
> [1] Datar, M., Immorlica, N., Indyk, P., and Mirrokni, V. S. Locality-sensitive hashing scheme based on p-stable distributions. In Proceedings of the twentieth annual symposium on Computational geometry, pp. 253–262, 2004.
>
> [2] Dasgupta, Anirban, Ravi Kumar, and Tamás Sarlós. Fast locality-sensitive hashing. In Proceedings of the 17th ACM SIGKDD international conference on Knowledge discovery and data mining. 2011.
>
> ## FastLSH Handles Sparse Data
> We are already aware of the issue you have raised. FastLSH can handle sparse data, as discussed in Appendix C.6. We use the MNIST dataset to validate this claim, and the results are shown in Table 9 in Appendix, where MNIST is a sparse dataset with around 2.6% non-zero elements. As you have focused on, for very sparse vectors, we can use the Hadamard transform to make the data dense. This makes FastLSH and AChash similar, so FastLSH can be considered a generalized version of AChash. In many practical applications, data is often dense, in which case FastLSH can be used directly. For sparse vectors, we apply the Hadamard transform for data preprocessing. This enhances the adaptability of FastLSH compared to AChash, enabling faster index construction and better query performance.

---

> ### Comment · Reviewer_ze7N · 2024-11-24
>
> I thank the authors for their response. I am still not convinced that the ideas in the paper add significant value over prior work. From what I understand, compared to ACHash, FastLSH doesn't have the Hadamard transformation step (which is crucial for such schemes to work in general), and in the second step, instead of subsampling the coordinates without replacement, it uses subsampling with replacement.
>
> At a conceptual level, the only place where the paper is suggesting something different is the subsampling scheme. Can the authors clarify if there is any significant conceptual advantage of subsampling with replacement, compared to without replacement as was done in ACHash? For example, are there reasonable settings, where one would expect the proposed subsampling scheme to do better and why?
>
> As far as the provable guarantees in the paper are concerned, the paper shows equivalence with LSH for m going to infinity, which defeats the whole purpose of subsampling. It is unclear if that is a meaningful contribution.

---

### Meta-Review · Area_Chair_pH5w · 2024-12-18

**Metareview:**

Thanks for your submission to ICLR.

The reviewers raised several concerns about the paper.  In particular, the first, third, and fourth reviewers provided concerns about the theoretical guarantees and prior work of the paper.  The authors responded to these concerns but the reviewers all followed up with further concerns, which were not addressed by the authors.  The second reviewer also had concerns about the empirical study.

Because the reviewers ultimately still have concerns with the paper after the discussion phase, and none of the reviewers are advocating for the paper to be accepted, I cannot recommend the paper to be published at this time.

**Additional Comments On Reviewer Discussion:**

The reviewers converged in agreement to not accept the paper.  They also responded to the author rebuttal with further questions which were not addressed by the authors.

---

### Decision · Program_Chairs · 2025-01-22

Reject